# Pharmacologic inhibition of PCBP2 biomolecular condensates relieves Alzheimer's disease

Lu Wang[1,2,3,4,5,10], Xiao-Yong Xie[1,2,3,4,5,10], Qiu-Ling Pan[1,2,3,4,5], Jiawei Zhang[6], Gui-Feng Zhou[1,2,3,4,5], Qi-Lei Zhang[7], Xiao-Xin Yan [7], Yu Xiang[1,2,3,4,5], Chen-Lu Li[1,2,3,4,5], Yi He[6], Xiao-Jiao Xiang[8], Xiao-Juan Deng[1,2,3,4,5], Yan-Jiang Wang[9], Ji-Ying Zhou[1,2,3,4,5], Shenyou Nie[6] & Guo-Jun Chen [1,2,3,4,5] ✉

Biomolecular condensates, membrane-less assemblies formed by phase separation, are implicated in neurodegenerative disease, but their role in Alzheimer's disease (AD) remains unclear. Here, we report that in the brain of AD patients and animal models, an elevation of poly(C)-binding protein 2 (PCBP2) correlates with biomolecular condensation that involves phase separation. These condensates sequester large numbers of mitochondrial and mRNA-binding proteins, leading to the outside impairment of mitochondrial morphology and function, and BACE1 mRNA decay relative to amyloid deposition. We then identify a small molecule CN-0928 that inhibits the condensates by reducing PCBP2 protein level and mitigates AD pathology and cognitive decline, in which CN-0928 binding to a target protein integrator complex subunit 1 (INTS1) allows to regulate PCBP2 expression. Our findings place PCBP2 condensates as a key player that cooperates the seemingly disparate but important pathways, and show pharmacological modulation of PCBP2 as an effective approach for treating AD.

Biomolecular condensates are micron-scale intracellular compartments without a surrounding membrane[1]. They concentrate proteins and nucleic acids through multivalent interactions[2,3]. In the cytoplasm, stress granule (SG) controls translation and cell signaling in response to stress[4]. These SGs recruit many RNA-binding proteins (RBPs) that condensate in neurodegenerative diseases[5,6]. It is proposed that chronic stress promotes the transfer of SG from transient structure to solid state, leading to aggregation of disease-associated RBPs[7,8]. Thus, condensate-modifying therapeutics have emerged as a novel strategy for drug discovery[9,10].

Extracellular amyloid deposition and intracellular Tau aggregation are two hallmarks of Alzheimer's disease (AD), with TDP-43 inclusions in about half of all cases[11,12]. Aggregation of Tau and TDP-43 involves liquid-liquid phase separation (LLPS)[13,14]. In contrast, little evidence supports that amyloid deposits directly involve phase separation. Amyloid protein (Aβ) is mainly localized to the membrane-surrounding multivesicular bodies along with the secretory pathway[15,16], where Aβ generation is regulated by APP processing enzymes, including BACE1, the rate-limiting protein[17]. These enzymes

[1]Department of Neurology, the First Affiliated Hospital of Chongqing Medical University, Chongqing, China. [2]Chongqing Key Laboratory of Major Neurological and Mental Disorders, the First Affiliated Hospital of Chongqing Medical University, Chongqing, China. [3]Chongqing Key Laboratory of Neurology, the First Affiliated Hospital of Chongqing Medical University, Chongqing, China. [4]Neurology Key Laboratory of Chongqing Education Commission of China, the First Affiliated Hospital of Chongqing Medical University, Chongqing, China. [5]Key Laboratory of Major Brain Disease and Aging Research (Ministry of Education), the First Affiliated Hospital of Chongqing Medical University, Chongqing, China. [6]Basic Medicine Research and Innovation Center for Novel Target and Therapeutic Intervention (Ministry of Education), College of Pharmacy, Chongqing Medical University, Chongqing, China. [7]Department of Anatomy and Neurobiology, Xiangya School of Medicine, Central South University, Changsha, Hunan Province, China. [8]Department of Nuclear Medicine, The Second Affiliated Hospital of Chongqing Medical University, Chongqing, China. [9]Department of Neurology and Centre for Clinical Neuroscience, Daping Hospital, Third Military Medical University, Chongqing, China. [10]These authors contributed equally: Lu Wang, Xiao-Yong Xie. ✉e-mail: 203165@cqmu.edu.cn

are translationally controlled by RBPs, as we have previously reported[18,19]. On the other hand, some RBPs tend to accumulate in condensate architecture, which tunes mRNA translation[20,21]. Protein-RNA binding may recruit a functional pool of regulatory factors, making them unavailable for normal functions[22,23]. It is thus likely that RBP-enriched granules may regulate amyloid deposition through a mechanism associated with RNA processing.

PCBP2 is an RBP that contains KH domains and specifically binds the cysteine-rich motif[24]. It is involved in cancer and iron metabolism[25]. Evidences also show PCBP2 reduces antiviral signaling[26] and regulates gene program in pancreatic cells[27]. PCBP2 transcript and protein levels are elevated in the brain of AD[28,29]. Despite that PCBP2 colocalizes with the processing body (PB) and is a constituent of SG[30,31], the functional role of PCBP2 in AD remains enigmatic.

Here, we show that PCBP2 forms condensates with biological functions relevant to AD, and we present a condensate-modifying compound with therapeutic potential.

## Results

### PCBP2 condensates are associated with elevated PCBP2 protein levels in AD

Although alteration of RBPs is the prominent feature and independent risk factor of AD[32,33], only a fraction of RBPs is presented in cytoplasmic granules[8]. Thus, identification of the potential RBPs should be important for understanding how SGs may contribute to the pathophysiology of AD[34]. We found that in the brain of AD patients, PCBP2 protein level was significantly increased compared with age-matched controls, and APP protein, which provides the only source for Aβ generation, was elevated accordingly (Fig. 1a and Supplementary Table 1). PCBP2 condensates that exhibited an irregular shape were significantly increased and enlarged in the cytoplasm of neurons (NeuN-positive) of AD relative to control (Fig. 1b). Similarly, significant enhancement of PCBP2 protein level correlated with neuronal PCBP2 condensation in both the cortex and hippocampus of 5×FAD mice (Fig. 1c–e and Supplementary Fig. 1a, b) and APP/PS1 mice (Supplementary Fig. 1c–e), two animal models of AD. Under normal condition, small and irregular PCBP2 granules were evenly distributed in nucleus of SH-SY5Y cells, whereas the enlarged PCBP2 clusters were localized to the cytoplasm, either in a cellular AD model of SH-SY5Y cells that stably overexpress APP (SH-SY5Y-APP), or in SH-SY5Y cells treated with SG inducer arsenite (Ars) (Fig. 1f). Interestingly, SH-SY5Y cells stably expressing mCherry-tagged PCBP2 (SH-SY5Y-mCherry-PCBP2) also exhibited a prominent condensation in the cytoplasm (Fig. 1g), indicating that high concentration of PCBP2 is sufficient for condensates. Elevation of PCBP2 protein level in these cells seemed to be within a physiological range, which was comparable to that in the brain of normal aging (Supplementary Fig. 1f–h). Moreover, PCBP2 clusters might be distinguishable from Aβ- or the microtubule-based aggresomes[35,36]. We showed that a majority of PCBP2 clusters did not colocalize with Aβ deposits in cells and brain sections of AD, as measured by Thioflavin S (ThioS) staining and the 6E10-positive immunosignals (Supplementary Fig. 1i–k). Similarly, in AD brain sections, HSP40-labeled aggresomes[35] rarely colocalized with PCBP2 clusters; partial overlap was detectable in cultured cells, where HSP40 staining was relatively uniform (Supplementary Fig. 1i–k). We further found that PCBP2 condensates underwent dynamic fusion in living cells (Supplementary Movie 1), and were inhibited by 1,6-hexanediol (1,6-HD) (Fig. 1h and Supplementary Movie 2), which is capable of dissolving other biomolecular condensates[37,38], suggesting a dynamic nature of the condensate assembly.

### PCBP2 undergoes LLPS

Given that PCBP2 condensates were promoted by the SG inducer Ars (Fig. 1f) and disrupted by 1,6-HD (Fig. 1h) in cultured cells, we examined the fusion behavior of purified full-length PCBP2 (PCBP2-A555) in an incubation buffer containing 10% PEG. At 5 μM, PCBP2-A555 formed liquid-like droplets that readily fused (Fig. 2a). Fluorescence recovery after photobleaching (FRAP) showed that fluorescence in the photobleached region of interest (ROI) recovered within 120 s (Fig. 2b). It is estimated that PCBP2 concentration is 1.28–1.40 μM under normal condition, based on that total cellular protein is about 5 mM[39,40], and that PCBP2 abundance in human brain is 256–279 ppm (parts per million) according to an online PaxDb data[41] [https://pax-db.org/protein/9606/ENSP00000352438]. This concentration of PCBP2 could be increased to 2 μM in the brain of aged mice and SH-SY5Y mCherry-PCBP2 cells (Supplementary Fig. 1f–h), where PCBP2 condensation is favored. Moreover, total RNA concentrations are approximately 150 ng/μL in human brain[42], and ~50 ng/μL in cultured human fibroblasts[43]. We next assessed PCBP2 condensation using PCBP2 at 0, 1, 2, and 4 μM, in combination with cell-extracted RNA at 0, 10, 20, and 50 ng/μL. We showed that PCBP2 began to form condensate at 2 μM in the presence of RNA at 10 ng/μL (Fig. 2c, d). With 2 μM PCBP2 and 20 ng/μL RNA, nascent droplets appeared within 5 min of incubation, grew and stabilized by ~10 min, and exhibited fusion by 1 h and 24 h (Supplementary Fig. 2a), indicating LLPS at physiologically relevant concentrations in vitro. In living SH-SY5Y cells, prominent mCherry-PCBP2 puncta were observed at 48 h but not 24 h post-transfection with the PCBP2 plasmid (Fig. 2e), and also in SH-SY5Y cells stably expressing mCherry-PCBP2 (Supplementary Fig. 2b). FRAP showed that fluorescence in condensate ROIs was rapidly lost upon photobleaching and recovered within ~60 s (Fig. 2e and Supplementary Fig. 2b). Together, these observations support that PCBP2 condensates form via LLPS.

### PCBP2 condensates recruited mitochondrial and RNA-binding proteins

To identify the components of PCBP2 condensates, fluorescence-activated particle sorting (FAPS) was used to purify mCherry-PCBP2 condensates that were stably expressed in SH-SY5Y and SH-SY5Y-APP cells (Fig. 3a and Supplementary Fig. 3a–e)[44]. An ideal isolation of mCherry-PCBP2 condensates was achieved (Supplementary Fig. 3a–e). Proteome analysis between the sorted and pre-sorted fractions in SH-SY5Y cells revealed 1209 proteins enriched in PCBP2 condensates (Welch's t-test, fold change >1.5 and P < 0.05). These proteins partially overlapped with p-body (PB) and SG components (Fig. 3b, c), and contained RNA-binding proteins that could be categorized into the 3′ untranslated region (3′UTR)-binding (PABPC4, FXR1 and ELAVL3), nonsense-mediated mRNA decay (NMD), (UPF1, SMG5, PABP1, EIF4A3, DCP1A, DCP1B, DCP2 and CASC3), miRNA pathway and RNA decapping complex (Supplementary Data 1 and 2). Only a few ribosomal proteins were enriched, suggesting that PCBP2 condensates did not sequester major translational machineries (Fig. 3d and Supplementary Data 1 and 2). Surprisingly, 140 mitochondrial components were significantly concentrated (Fig. 3e), ranging from respiratory chain complexes (NDUFV1, NDUFS1, and ND2), ATP synthesis proteins (ATP5F1D, ATP5F1B, ATP5PD), to translocases (TOM40/6 and TIMM23/17B) (Supplementary Data 1). Gene ontology analysis revealed extra proteins associated with synaptic signaling and response to Aβ, in addition to mitochondrial and RNA-binding proteins (Fig. 3f and Supplementary Data 2), which were relevant to molecular mechanisms of AD. Similar proteins and pathways were also enriched by PCBP2 condensates in SH-SY5Y-APP cells (Supplementary Fig. 4a–e and Supplementary Data 3), suggesting that high concentrations of PCBP2 were able to form relatively stable aggregates, irrespective of intracellular APP/Aβ levels. We further showed that example proteins, including DCP1a and DDX6 found in PB, TIA1 in SG, and TOM20 in mitochondria, were colocalized with mCherry-PCBP2 (Fig. 3g), indicating an effective sequestration. The mitochondrial protein TOM20 was then selected to identify whether this is a client protein for PCBP2, given that no IDR is presented in the protein sequence (uniport number Q15388). Physiological concentration of TOM20 is estimated to be 0.22-0.69 μM[39,41]. We showed that different combination of purified TOM20 (TOM20-D488 at 0.5 or 1 μM) with 10% PEG or 50 ng/

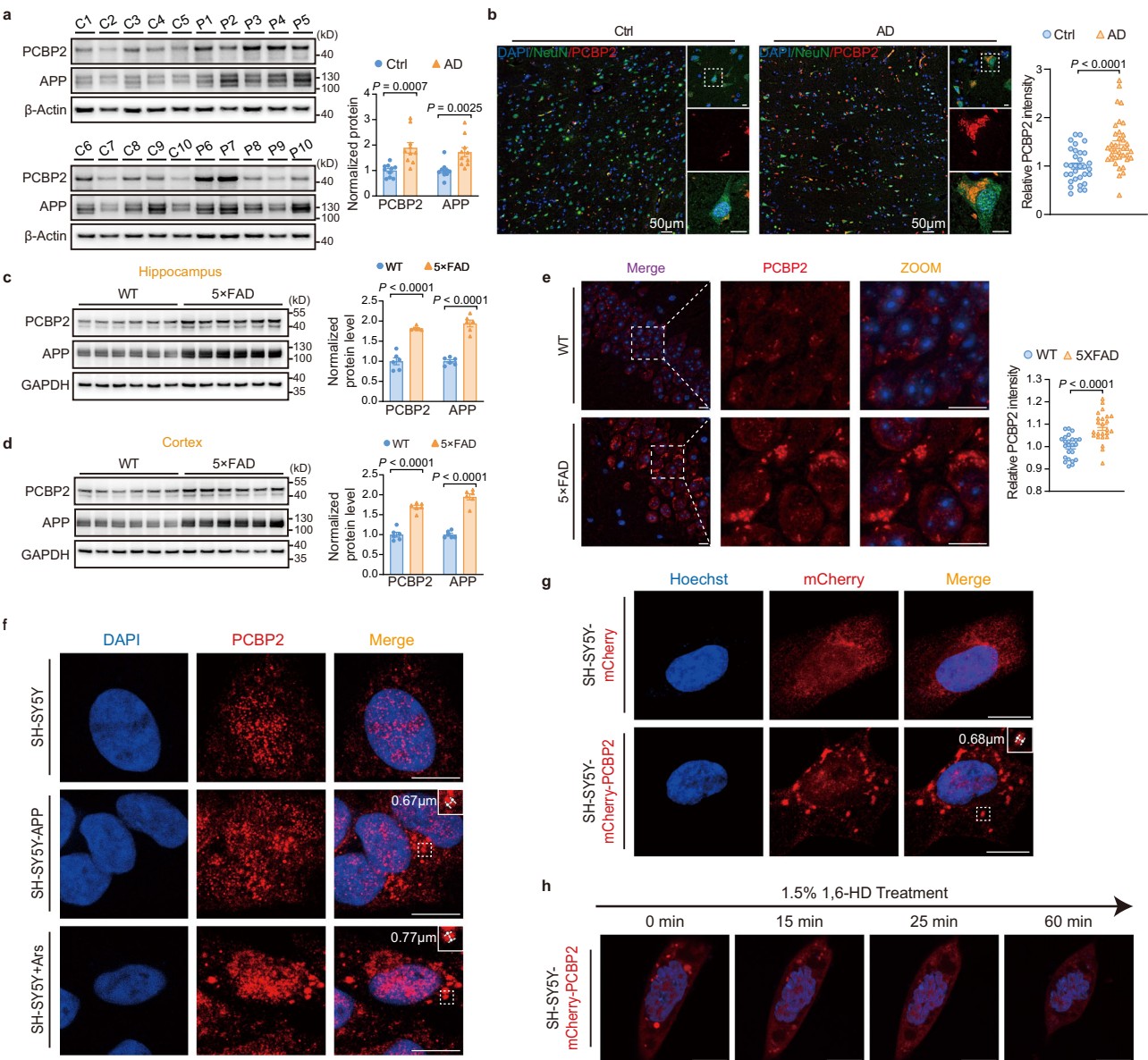

**Fig. 1 | PCBP2 condensates correlate with increased PCBP2 protein abundance in AD. a** Western blots (left) and quantification (right) of PCBP2 and APP in the temporal cortex of postmortem brains from both non-AD subjects (Ctrl, C1-C10) and AD patients (AD, P1-P10). **b** Representative immunofluorescence images (left) and quantification (right) of PCBP2 condensates (red) in the temporal cortex from non-AD controls and AD patients. NeuN (green) and DAPI (blue) are counterstained. Scale bar, 10 μm or as indicated. **c, d** Western blots (left) and quantification (right) of PCBP2 and APP protein levels in the hippocampus (**c**) and cortex (**d**) of 5×FAD and wild-type (WT) mice. **e** Representative immuno-fluorescence images (left) and quantification (right) showing PCBP2 condensates (red) in the hippocampus of 12-month-old WT and 5×FAD mice. Scale bar, 10 μm.

**f** Representative immunofluorescent images of PCBP2 (red) in SH-SY5Y, SH-SY5Y-APP, or SH-SY5Y cells treated with arsenite (Ars, 0.5 mM for 30 min). Scale bar, 10 μm. **g** Live-cell imaging of SH-SY5Y cells stably expressing mCherry-PCBP2 (SH-SY5Y-mCherry-PCBP2), counterstained with Hoechst (blue). Scale bar, 10 μm. **h** Time-lapse confocal fluorescence imaging of SH-SY5Y cells stably expressing mCherry or mCherry-PCBP2, following treatment with 1.5% 1,6-hexanediol (1,6-HD). Scale bar, 10 μm. Data are expressed as mean ± s.e.m (**a**–**e**). Significance was determined by the unpaired two-tailed Student's $t$-test (**a**–**e**). In vitro: $n = 3$ biological replicates (**f**–**h**). In vivo: $n = 6$ mice/group (**c**–**e**); $n = 7$ samples (**b** [Ctrl group]); $n = 8$ samples (**b** [AD group]); $n = 10$ samples (**a**). Source data are provided as a Source Data file.

μL RNA did not lead to a droplet formation, whereas co-incubation of TOM20-D488 (0.5 μM) with PCBP2 (PCBP2-A555, 2 μM) in the presence of RNA (20 ng/μL) yielded a clear liquid-like droplet (Fig. 3h). These results indicate that mitochondrial and RNA-binding proteins are the major components of PCBP2 condensates.

### Disrupted mitochondrial morphology and function

Gene set enrichment analysis (GSEA) demonstrated that mitochondrial proteins were significantly enriched in PCBP2 condensates (Fig. 4a). These proteins, sequestered by PCBP2, are not similarly found in reported PB or SG, and may have an impact on mitochondrial assembly

and function[45]. Immunofluorescence imaging stained with mito-tracker and TOM20 showed that mitochondrial morphology was sig-nificantly impaired in mCherry-PCBP2 cells, as measured by area, perimeter, aspect ratio (length-to-width), and form factor (complexity and branching) values (Fig. 4b, c). Transmission electron microscopy (TEM) images showed that the mitochondria were damaged with ridge reduction or absence (Fig. 4d). Sequestration of respiratory chain and ATP synthesis proteins by PCBP2 condensates might impair ROS and ATP homeostasis[46,47]. We further showed that ROS levels were sig-nificantly increased in mCherry-PCBP2 cells (Fig. 4e), with a significant reduction in oxygen consumption measured by respiration rate and

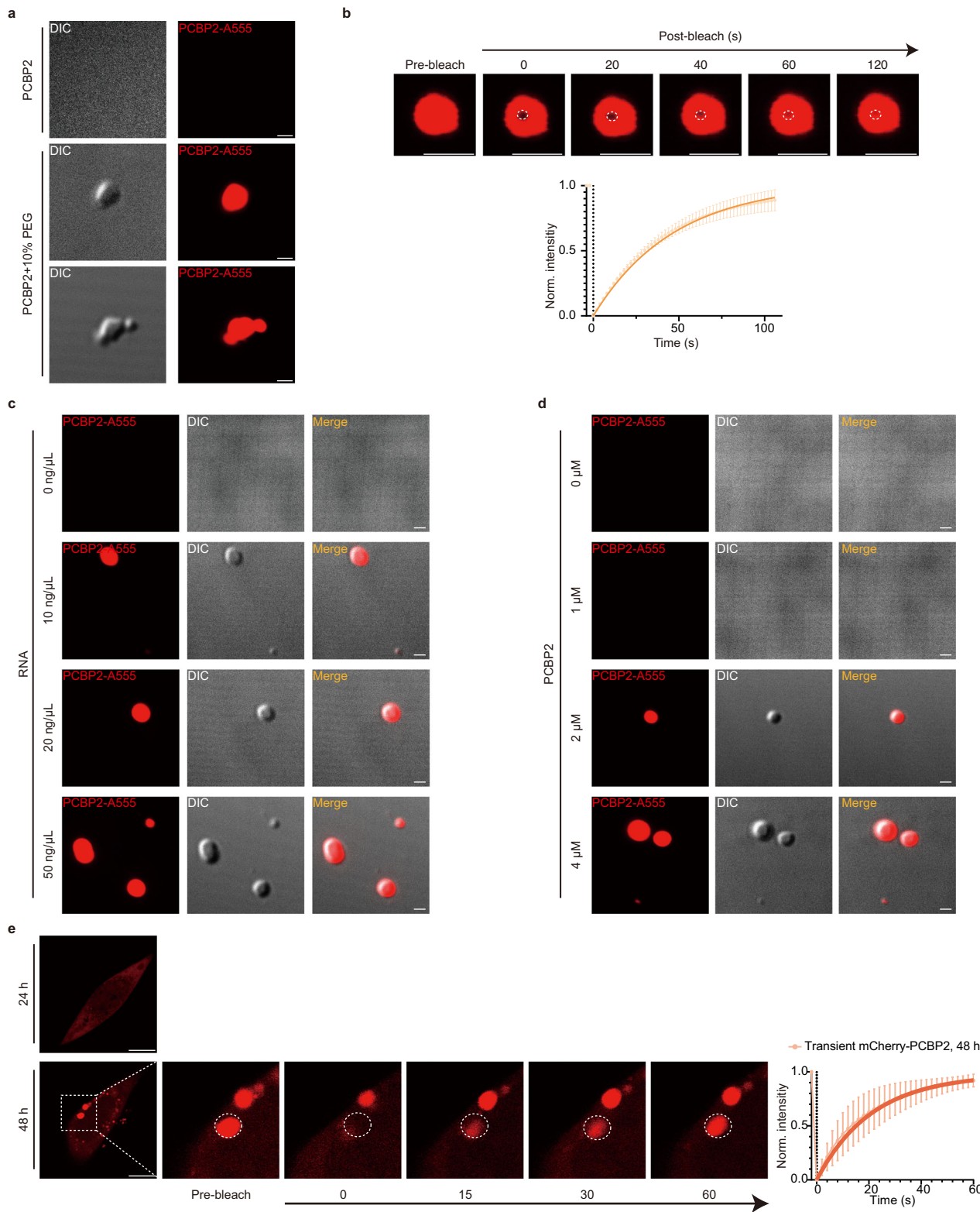

**Fig. 2 | PCBP2 forms liquid-like condensates via LLPS. a** Phase separation of the full-length PCBP2 (PCBP2-A555, 5 μM) occurs in 10% PEG, forming liquid droplets that are visualized by the differential interference contrast (DIC) channel and the fluorescent channel of confocal microscopy. The middle panel shows an isolated single small condensate, whereas the bottom panel shows a cluster of condensates. Scale bar, 1 μm. **b** Fluorescence recovery after photobleaching (FRAP) of PCBP2-A555 droplets was assessed in 10% PEG. The dashed box indicates the photobleached region (region of interest, ROI). Scale bar, 2.5 μm. **c** Representative confocal images of liquid-like droplet formation by PCBP2 in vitro at fixed PCBP2

(2 μM) with RNA at 0, 10, 20, and 50 ng/μL. Scale bar, 1 μm. **d** Representative confocal images at fixed RNA (20 ng/μL) with PCBP2 at 0, 1, 2, and 4 μM. Scale bar, 1 μm. **e** mCherry-PCBP2 condensates are observed in SH-SY5Y cells at 48 h post-transfection but not at 24 h. Representative fluorescence images (left) and FRAP recovery curve (right) at 48 h illustrate fluorescence recovery within condensates; the dashed box marks the photobleached region of interest (ROI). Scale bar, 10 μm. Data are presented as mean ± SD (**b**, **e**). In vitro: *n* = 3 biological replicates (**a–e**). Source data are provided as a Source Data file.

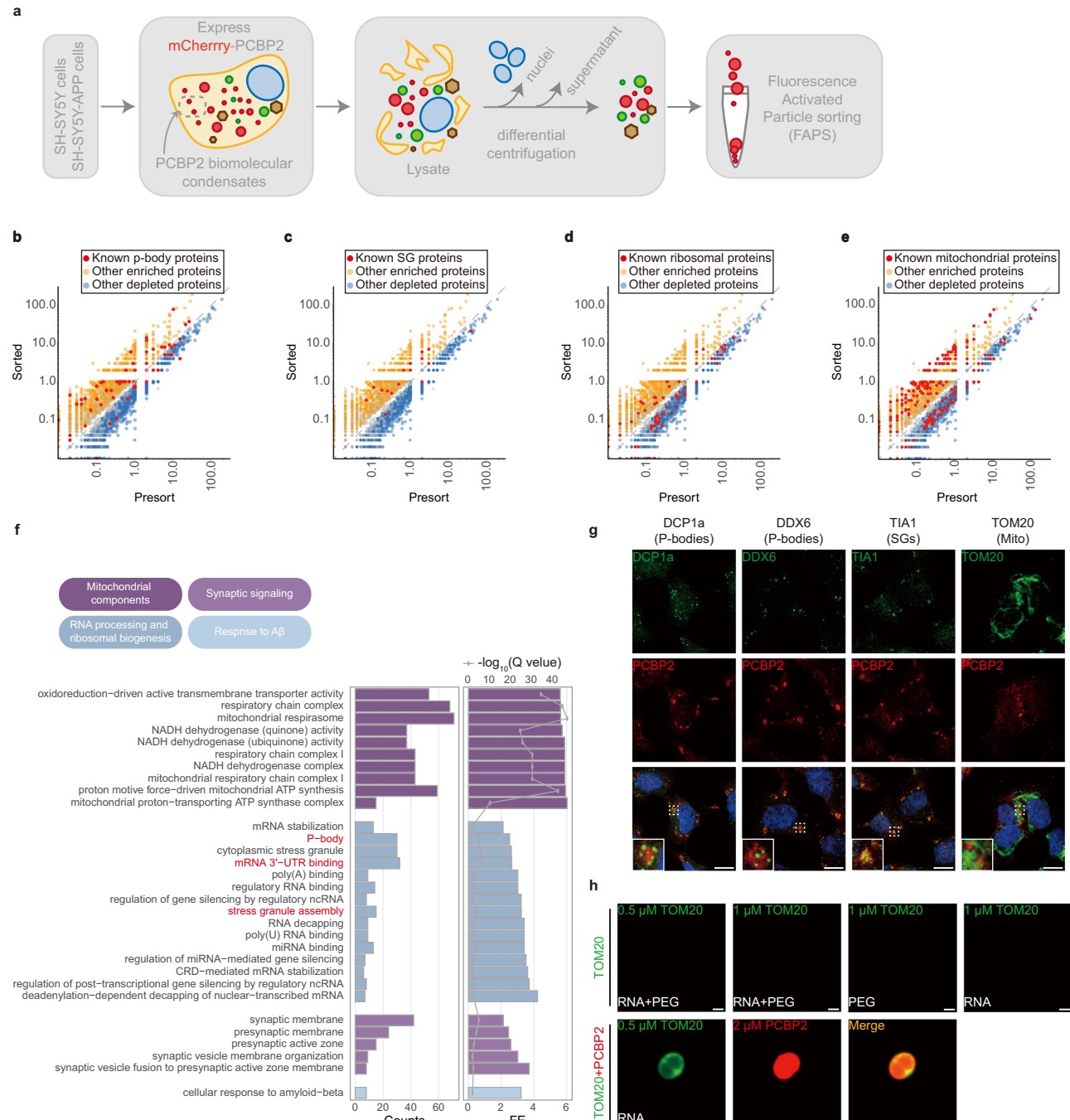

**Fig. 3 | PCBP2 condensates sequester mitochondrial and RNA-binding proteins.** **a** Isolation and analysis workflow for the characterization of PCBP2 condensates by fluorescence-activated particle sorting (FAPS). **b–e** Comparison of protein abundances between sorted and pre-sorted fractions utilizing quantitative values from normalized total spectra of LC-MS/MS. Significantly enriched and depleted proteins are indicated, alongside known p-body proteins (**b**), stress granule (SG) proteins (**c**), ribosomal proteins (**d**), and mitochondrial proteins (**e**). **f** Proteins identified by mass spectrometry were analyzed by gene ontology using the clusterProfiler with associated protein numbers (counts), fold enrichment (FE) values, and Q-values. **g** Representative images of mCherry-PCBP2 condensates colocalized with the indicated organelle-associated proteins by confocal sectioning. Scale bar, 10 μm. **h** No droplet is found when purified recombinant human TOM20-D488 (0.5 or 1 μM) is incubated with 10% PEG or 50 ng/μL RNA as indicated. Co-incubation of purified PCBP2-A555 (2 μM) with TOM20-D488 (0.5 μM) in buffers containing 20 ng/μL RNA yields a fluorescent droplet. Scale bar, 1 μm. In vitro: $n$ = 3 biological replicates (**g**, **h**). Source data are provided as a Source Data file.

ATP levels, and extracellular acidification rate by glycolysis and glycolysis capacity, respectively (Fig. 4f, g). Also, as shown in Fig. 4h, i, the colocalization of PCBP2 condensates with the granulated TOM20 was significantly enhanced in SH-SY5Y-APP cells compared to SH-SY5Y cells; and silencing of PCBP2 in SH-SY5Y-APP cells led to a significant reduction of ROS level. These results indicate that PCBP2 condensates are directly associated with the damaged mitochondrial structure and function, which connects the well-documented synaptic and cognitive impairment in AD[48,49].

## Impaired BACE1 3'UTR degradation relative to amyloid deposition

Sequestration of the 3'UTR-binding proteins suggested that PCBP2 condensates might regulate mRNA abundance in the cytoplasm[50]. RNA

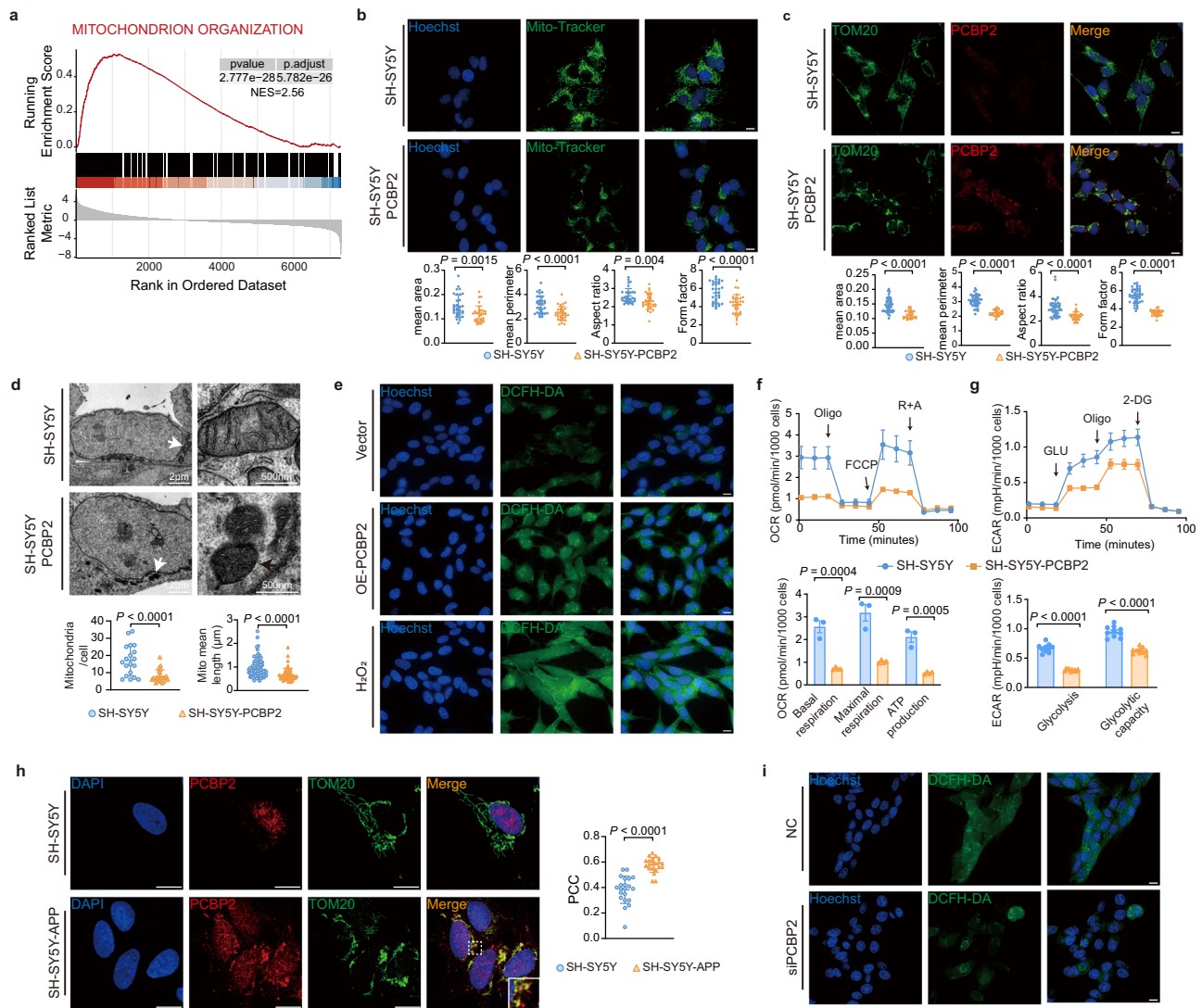

**Fig. 4 | PCBP2 condensates disrupt mitochondrial morphology and function.**
**a** GSEA comparing pre- and post-sorted PCBP2-condensate fractions used a one-sided enrichment test; Multiple testing was controlled across gene sets using the Benjamini–Hochberg FDR. **b** Live-cell imaging (upper) of SH-SY5Y-PCBP2 cells stained with MitoTracker Green. Quantitative analysis (lower) of mitochondrial morphology is measured by area, perimeter, aspect ratio (AR), and form factor (FF) values using a computer-assisted morphometric system. Scale bar, 10 μm.
**c** Immunofluorescence images (upper) of SH-SY5Y-PCBP2 cells stained for TOM20 (green), DAPI (blue), and PCBP2 (red). These descriptors (lower) also included area, perimeter, AR, and FF. Scale bar, 10 μm. **d** Transmission electron microscopy (TEM) images show an impairment of mitochondrial morphology in SH-SY5Y-PCBP2 cells. Mitochondria indicated by the white arrows in the left panels are displayed with higher magnification in the right panels. Black arrows indicate damaged mitochondria with ridge reduction or absence. Scale bar, 2 μm or as indicated. Mitochondrial (mito) mean number and length were quantified. **e** Live-cell imaging of reactive oxygen species (ROS) levels in SH-SY5Y cells transiently transfected with

vector (Vector), PCBP2, or incubated with 50 μM hydrogen peroxide ($H_2O_2$) for 4 h without transfection. ROS was labeled with the DCFH-DA probe. Scale bar, 10 μm.
**f** Oxygen consumption rate (OCR) between SH-SY5Y and SH-SY5Y-PCBP2 cells is assessed using a Seahorse XF24 Extracellular Flux Analyzer, respectively. Statistical results are presented as bar graphs at the bottom. **g** Extracellular acidification rate (ECAR) between SH-SY5Y and SH-SY5Y-PCBP2 cells is assessed using a Seahorse XF24 Extracellular Flux Analyzer, respectively. Statistical results are presented as bar graphs at the bottom. **h** Immunofluorescence images showing the colocalization of TOM20 (green) and PCBP2 (red) in SH-SY5Y-APP cells. Scale bar, 10 μm. PCC: Pearson's correlation coefficient. **i** Live-cell imaging of reactive oxygen species (ROS) levels in SH-SY5Y-APP cells transiently transfected with siPCBP2 or a negative control (NC). ROS levels were visualized with the DCFH-DA probe. Scale bar, 10 μm. Data: mean ± SD (**b**–**d**, **f**–**h**) and analyzed by two-tailed Student's $t$-test (**b**–**d**, **f**–**h**). In vitro: $n = 3$ biological replicates (**b**–**e**, **f** [SH-SY5Y], **h**, **i**); $n = 4$ biological replicates (**f** [SH-SY5Y-PCBP2]); $n = 10$ biological replicates (**g**). Source data are provided as a Source Data file.

sequencing was employed to identify differentially expressed genes (DEGs) by PCBP2 knockdown in SH-SY5Y cells. Silencing of PCBP2 led to a total of 416 DEGs, including 171 upregulated and 245 down-regulated genes (Fig. 5a). BACE1 mRNA level was among those significantly decreased (Fig. 5a), which correlated with a significant reduction of protein levels of BACE1 and β/α C-terminal fragments of APP (CTFs) (Fig. 5b and Supplementary Fig. 5b). Conversely, PCBP2 overexpression augmented protein levels of CTFs (Supplementary Fig. 5a, c). PCBP2 silencing, as opposed to overexpression, led to

significantly lower levels of Aβ40/42 in two cell lines stably expressing APP (Supplementary Fig. 5d,e). We further showed that PCBP2 silencing shortened the half-life of BACE1 mRNA, and accordingly, the 3′ UTR but not 5′UTR activity was significantly enhanced by PCBP2 overexpression (Fig. 5c, d). These results indicate that PCBP2 controls the 3′UTR-dependent BACE1 mRNA stability and downstream amyloid deposition.

To determine the 3′UTR interaction with potential RBPs, the 5-bromo-UTP (BrU)-labeled 3′UTR was first in vitro synthesized and

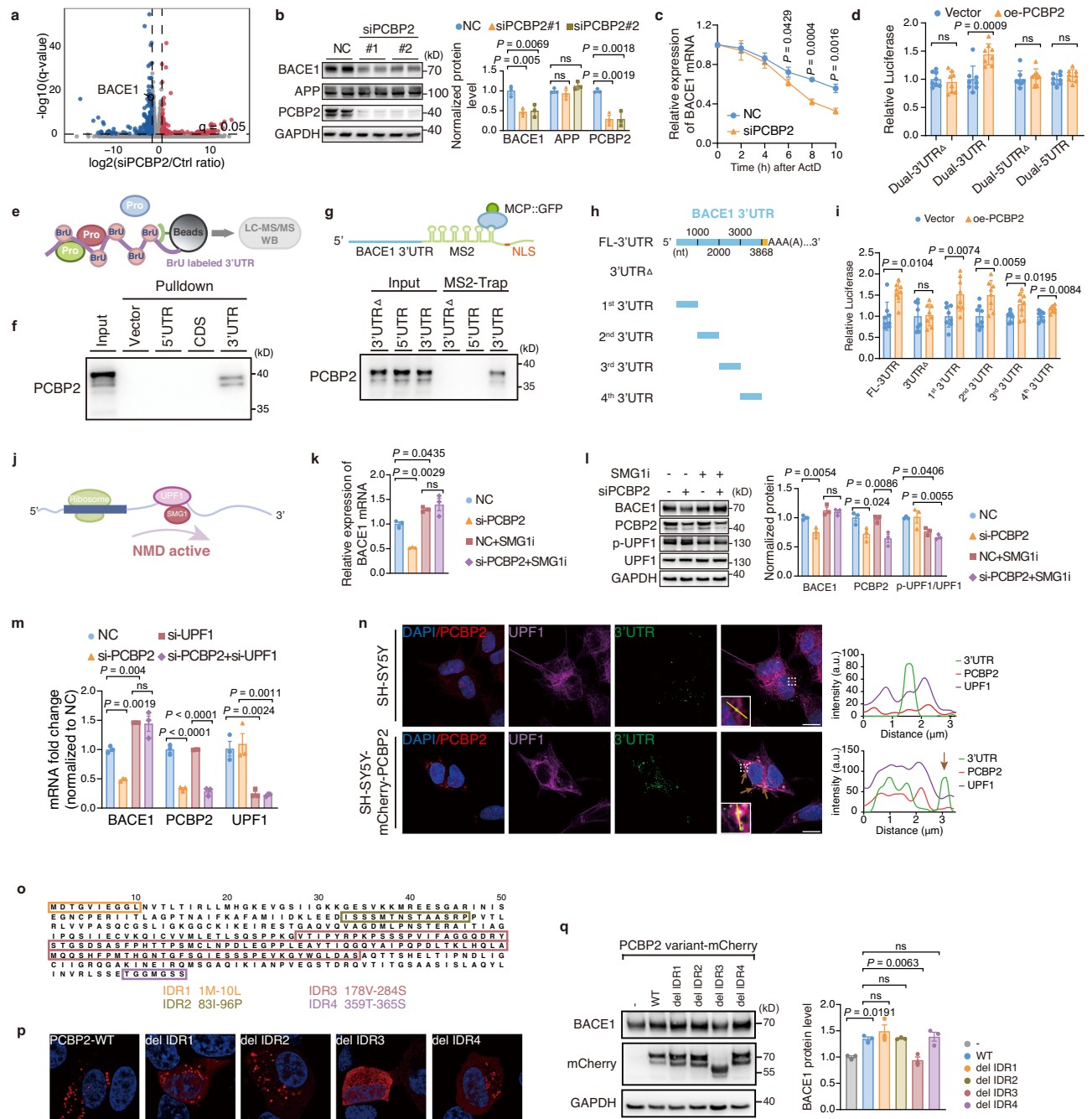

**Fig. 5 | PCBP2 condensates inhibit the 3′UTR degradation of BACE1 mRNA.**
**a** Scatter plot of differentially expressed genes in SH-SY5Y cells shows BACE1
transcript is downregulated by PCBP2 knockdown. **b** Western blots and quantifi-
cation indicate that BACE1 protein is reduced by PCBP2 knockdown. **c** Actinomycin-
D chase (0–10 h) shows faster decay of BACE1 mRNA with siPCBP2. **d** Dual-
luciferase assays 48 h post-transfection with overexpression PCBP2 (oe-PCBP2)
and reporters with or without 3′/5′UTRs reveal repression through the 3′UTR.
**e** Schematic diagram of BrU-labeled 3′UTR RNA pulldown followed by LC-MS/MS to
identify 3′UTR-binding RBPs. **f** Western blots of PCBP2 in immunoprecipitated
extracts by RNA (BACE1 3′UTR) pulldown. **g** MS2-Trap and western blot confirm
PCBP2 associates with the 3′UTR but not the 5′UTR; 3′UTRΔ, deletion control.
**h** Map of the full-length BACE1 3′UTR truncated into four segments for reporter
testing. **i** Relative luciferase activities of the truncated 3′UTRs reporter plasmids.
**j** Schematic illustration shows nonsense-mediated mRNA decay (NMD) involves
UPF1 and SMG1. **k**, **l** siPCBP2 lowers BACE1 mRNA (**k**) and protein (**l**); SMG1 inhibitor
(SMG1i, 300 nM, 4 h) mitigates these effects; p-UPF1/UPF1 blots shown. **m** UPF1

knockdown rescues the siPCBP2-induced reduction in BACE1 mRNA.
**n** Immunofluorescence of SH-SY5Y and SH-SY5Y-PCBP2 cells shows colocalization
of PCBP2 (red), UPF1 (magenta), and BACE1 3′UTR (green, dEcCas6 probe); line
scans along the yellow ROI; brown arrows indicate regions where the 3′UTR is
localized with low UPF1 density. Scale bar: 10 μm. **o** PCBP2 domain map high-
lighting four predicted intrinsically disordered regions (IDR1-IDR4). **p** Confocal
images of PCBP2-mCherry clusters formed by the indicated deletion mutants
lacking IDR1-IDR4 (del IDR1-4) transiently expressed in SH-SY5Y cells. Scale bar,
2.5 μm. **q** Representative western blots and corresponding quantification of BACE1
and the PCBP2-mCherry deletion mutants (del IDR1-4) in SH-SY5Y cells. Data:
mean ± s.e.m (**b**, **k**–**m**, **q**) and mean ± SD (**c**, **d**, **i**). Significance was determined by the
unpaired two-tailed Student's t-test (**c**, **d**, **i**), one-way ANOVA with Bonferroni's test
(**b**, **k**–**m**, **q**). ns: nonsignificant. In vitro: n = 3 biological replicates
(**b**, **c**, **f**, **g**, **k**–**n**, **p**, **q**); n = 7 biological replicates (**i** [4th-3′UTR oe-PCBP2 group]); n = 8
biological replicates (**d**, **i** [other groups]). Source data are provided as a Source
Data file.

mixed with cell lysates for subsequent LC-MS/MS analysis (Fig. 5e)[18]. PCBP2, along with other RBPs, including FXR1 and STAU1, which show BACE1 mRNA-binding property as we have previously reported[18,51], was identified (Supplementary Data 4). RNA pulldown and MS2-trap assay further showed that PCBP2 specifically bound to the 3′UTR but not the 5′UTR of BACE1 (Fig. 5f, g). A dEcCas6-based fluorescence-activated RNA probe showed that the 3′UTR colocalized with the clustered and surrounding non-clustered fractions of mCherry-PCBP2 (Supplementary Fig. 6a–c). This pattern was similarly provoked by the SG inducer Ars in SH-SY5Y cells (Supplementary Fig. 6d). The 3′UTR binding was dependent on the KH2 domain of PCBP2, as revealed by RNA immunoprecipitation (RIP) combined with RT-qPCR analysis (Supplementary Fig. 6e–g). These results indicate a KH2-dependent PCBP2 binding to the 3′UTR of BACE1 mRNA.

The full-length 3′UTR was then cut into four fragments to screen for involvement of potential sequence/motif. PCBP2 overexpression nonspecifically enhanced the luciferase activity of all of these fragments (Fig. 5h, i). This was reminiscent of nonsense-mediated mRNA decay (NMD) mechanism, in which the long 3′UTR favors the binding of an RNA helicase UPF1 in a sequence-independent manner[52]. The key event in NMD is the UPF1 phosphorylation by the RNase SMG1 (Fig. 5j)[53,54]. We showed that the SMG1 inhibitor significantly increased BACE1 mRNA and prevented the effect of PCBP2 on BACE1 mRNA and protein levels, correlating with a reduced protein level of phosphorylated UPF1 (p-UPF1) (Fig. 5k, l). UPF1 knockdown also led to a significant enhancement of BACE1 mRNA and further diminished PCBP2-mediated regulation of BACE1 mRNA (Fig. 5m). Unexpectedly, we found that PCBP2 is able to recruit both UPF1 (Supplementary Data 2) and the BACE1 3′UTR. This dual interaction would normally be expected to accelerate the decay of BACE1 mRNA; however, our data revealed that PCBP2 instead stabilizes the transcript. These findings prompted us to further explore how PCBP2 condensates regulate BACE1 mRNA. We therefore assessed the spatial relationship among UPF1, PCBP2 condensates, and the BACE1 3′UTR, which was visualized with a dEcCas6-based fluorescent probe. As shown in Fig. 5n, UPF1 was markedly enriched within PCBP2 condensates, and the overall 3′UTR signal was significantly higher in SH-SY5Y-mCherry-PCBP2 cells than in parental SH-SY5Y cells. Colocalization analysis confirmed that both the BACE1 3′UTR and UPF1 are recruited into PCBP2 condensates, consistent with our previous findings. Nevertheless, a substantial fraction of 3′UTR transcripts remained outside the condensates, and these extra-condensate molecules were preferentially located in regions with low UPF1 density (brown arrowheads in Fig. 5n). These observations suggest a plausible mechanism: by sequestering UPF1 inside the condensates, PCBP2 reduces the degradation of 3′UTR transcripts that remain outside them.

To determine whether the regulation of BACE1 mRNA decay was dependent on PCBP2 condensation, we mutated PCBP2 with different fragments being deleted based on an online software DISOPRED3 that predicts the potential intrinsically disordered region (IDR) (Fig. 5o). Compared to other PCBP2 mutants that were equally transfected under same condition, del-IDR3 disrupted condensate formation (Fig. 5p), and significantly reduced BACE1 protein level (Fig. 5q). Thus, sequestration of NMD machinery by PCBP2 condensates could impair the 3′UTR decay, leading to an enhanced BACE1 mRNA and protein level in association with amyloid deposition.

## Small-molecule CN-0928 reduced condensates and alleviated AD

High concentration-dependent formation of PCBP2 condensates provides an opportunity to explore the potential therapeutic intervention by simply reducing PCBP2 protein level. Compounds in CNS MPO library from Topscience were filtered by backbone diversity, leading to final 3000 small molecules showing blood-brain barrier-penetrating potential (Supplementary Table 2)[55]. Through live-cell image screening

using mCherry-PCBP2, we identified that CN-0928 significantly reduced PCBP2 protein level, with the concomitant reduction of BACE1 but not APP or ADAM10 protein level (Fig. 6a, b). Dose-response effect showed that the effective concentrations ranged from 1 to 2 μM without obvious toxicity (Supplementary Fig. 7a, b), and the time-course response revealed that the BACE1-reducing effect of CN-0928 (1 μM) lasted for up to 96 h (Supplementary Fig. 7c). Accordingly, PCBP2 condensates in SH-SY5Y-mCherry-PCBP2 cells were significantly reduced by CN-0928 (Fig. 6c). In line with the function of PCBP2 condensates, the catalytic products of BACE1 measured by protein levels of CTFs and Aβ40/42 were significantly reduced (Fig. 6d, e), whereas morphological impairment of mitochondria significantly attenuated in SH-SY5Y cells (Supplementary Fig. 7d).

In an animal model of 5×FAD mice, intraperitoneal injection of CN-0928 significantly reduced protein levels of PCBP2 and BACE1 in the brain (Fig. 6f, g), with a significant reduction of PCBP2 condensates (Fig. 6h). Protein levels of CTFs as well as Aβ40/42 immunofluorescent signals and proteins in 5×FAD mice were significantly reduced by CN-0928 (Fig. 6i–l). An improvement of cognitive functions in CN-0928-treated mice was demonstrated by the significantly reduced escape latency and prolonged target quadrant passing and staying times in Morris water testing (Fig. 6m–p).

## CN-0928 downregulated PCBP2 protein by INTS1

We labeled CN-0928 with biotin, which allowed the following streptavidin to probe CN-0928. Through a series of chemical reactions, HCl was first introduced to generate CN-0928-HCl, before the final synthesis of CN-0928-biotin (Supplementary Fig. 8a-d). In comparison with CN-0928-HCl, which significantly decreased PCBP2 and downstream BACE1 protein levels in SH-SY5Y cells, CN-0928-biotin did not cause a significant alteration of PCBP2 protein level (Supplementary Fig. 9a, b). Since the modification sites remained unchanged, this difference is likely attributable to membrane permeability. The effect of CN-0928-HCl was further confirmed in 5×FAD mice, which exhibited a significantly reduced amyloidogenesis (Supplementary Fig. 9c–f) and improved cognitive function (Supplementary Fig. 9g–j). Chemoproteomics was then applied to target protein screening of PCBP2 by using CN-0928-biotin as a probe (Fig. 7a)[56,57]. This led to a total of 6 candidate proteins (mH2A1, INTS1, CKM, TOMM20, TUBGCP2, DNAJC16) (Supplementary Data 5), whereas ligandability analysis showed that more proteins might be involved in CN-0928 binding (Supplementary Fig. 10a and Supplementary Data 5). Core histone macro-H2A.1 (mH2A1) and integrator complex subunit 1 (INTS1) proteins that showed high relevance were selected for further verification. Silencing of INTS1 but not H2A.1 mediated the effect of CN-0928 on PCBP2 protein level (Fig. 7b,c). Moreover, except CKM, which was undetectable in SH-SY5Y cells [https://www.proteinatlas.org], knockdown of other proteins, TOMM20, TUBGCP2, and DNAJC16, did not prevent CN-0928-mediated reduction of PCBP2 and downstream BACE1 protein (Supplementary Fig. 10b–d), indicating that INTS1 is the principal mediator. Silencing of INTS1 also significantly reduced PCBP2 mRNA level and further attenuated the effect of CN-0928 on PCBP2 mRNA (Fig. 7d), suggesting that INTS1 controlled PCBP2 expression through a transcriptional mechanism. A computerized molecular docking model showed that INTS1 bound to CN-0928 with high affinity, with Arg-1404 being the potential binding site (Fig. 7e, f). We finally showed that mutations of Arg-1404 in INTS1 (R1404A, R1404L, and R1404K) prevented the binding of CN-0928-biotin to flag-labeled INTS1 (Fig. 7g). These data indicate that CN-0928 binds to INTS1 through R1404, which regulates PCBP2 transcription and protein level.

## Discussion

Identifying the key molecules within protein aggregates that drive AD progression remains challenging, as therapeutic approaches targeting the deposition of Aβ and tau fail most clinical trials. Large-scale

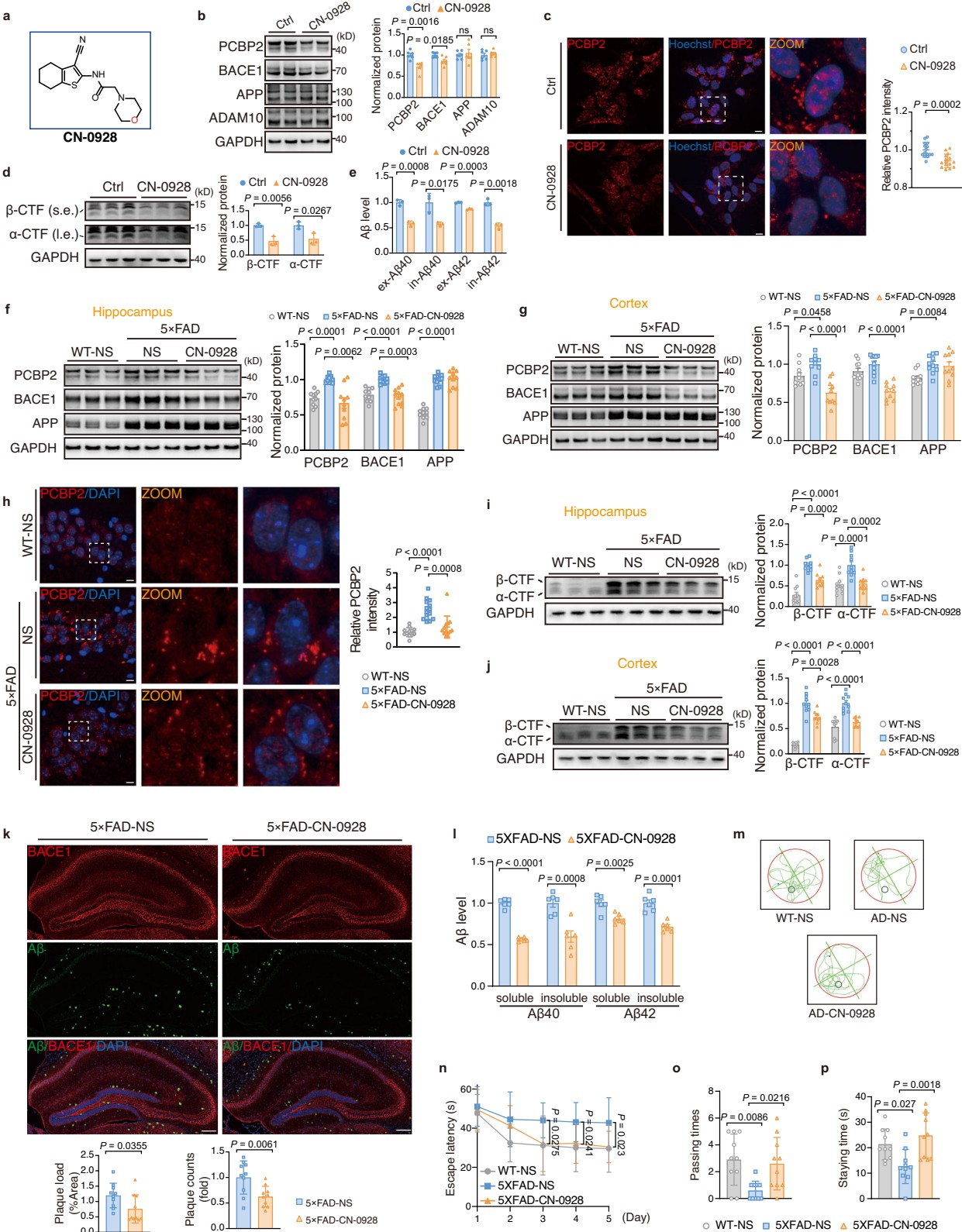

proteomics study provides a clue that alteration of RBPs is prominent[32]. Some RBPs, including snRNP70 (U1-70K) and HNRNPA2B1, are also enriched in the insoluble fractions of AD brains[58,59]. U1-70K has been shown to self-assemble and to aggregate with tau and other proteins implicated in AD[60], whereas HNRNPA2B1 is a component of the tau interactome and mediates the progression of tauopathy[61,62]. Notably, these proteins undergo phase separation[63,64], supporting that

biomolecular condensates contribute to protein aggregation[22]. In our study, we demonstrate that PCBP2 condensation represents a pathological feature that may serve as a disease driver by regulating mitochondrial dysfunction and amyloidogenesis in AD. As PCBP2 condensation is exaggerated either by the SG inducer Ars or in the aged brain, where PCBP2 protein level is elevated, we speculate that cellular stress and aging may work cooperatively for condensate formation[8].

**Fig. 6 | Small-molecule CN-0928 inhibits condensates by reducing PCBP2 levels and alleviates AD. a** Chemical Structure of CN-0928. **b–e** Cells were treated with either DMSO or CN-0928 (1 μM) for 48 h. **b** Western blots (left) and quantification (right) of PCBP2, BACE1, APP, and ADAM10 in SH-SY5Y cells treated with CN-0928. **c** Live-cell imaging (left) and quantification (right) of PCBP2 condensates in SH-SY5Y-mCherry-PCBP2 cells incubated with CN-0928. Scale bar: 10 μm. **d** Western blots (left) and quantification (right) of α/β-CTFs. s.e. short exposure; l.e., long exposure. **e** Aβ40 and Aβ42 levels in cell lysates (in) and culture medium (ex) were measured by ELISA. **f–p** Male 5×FAD mice were injected with either normal saline (NS) or CN-0928. Wild-type (WT) mice received NS as the complete blank control. **f, g** Representative western blots (left) and quantification (right) of PCBP2, BACE1, and APP levels in the hippocampus (**f**) and cortex (**g**). **h** Representative immuno-fluorescence images (left) and quantification (right) of PCBP2 condensates in the hippocampus. Scale bars, 10 μm. **i, j** Representative western blots (left) and

quantification (right) of α/β-CTFs levels in the hippocampus (**i**) and cortex (**j**). **k** Immunofluorescent labeling (upper) and quantification (lower) of Aβ deposits (green) in the hippocampus. Scale bar: 200 μm. **l** Soluble and insoluble Aβ40 and Aβ42 in the hippocampus were quantified using ELISA. **m** Representative trajectory maps for three groups of mice in the probe trials. **n** 5×FAD mice treated with CN-0928 exhibit a shorter escape latency compared with NS. **o, p** Passing times (**o**) and the staying time (**p**) in the target quadrant in different groups of mice. Data: mean ± s.e.m (**b, f, g, i, j, l**) and mean ± SD (**c, h, k, n–p**). Significance was determined by the unpaired two-tailed Student's *t*-test (**b–e, k, l**), one-way ANOVA with Bonferroni's test (**g, i, j** [α-CTF], **n, p**), Welch's one-way ANOVA with Dunnett's T3 post hoc test; Brown-Forsythe variance test (**f, j** [β-CTF], **o**), and Kruskal–Wallis test with Dunn's post-test (**h**). ns: nonsignificant. In vitro: *n* = 3 biological replicates (**c–e**); *n* = 6 biological replicates (**b**). In vivo: *n* = 6 mice/group (**h, l**); *n* = 10 mice/group (**f, g, i–k, n–p**). Source data are provided as a Source Data file.

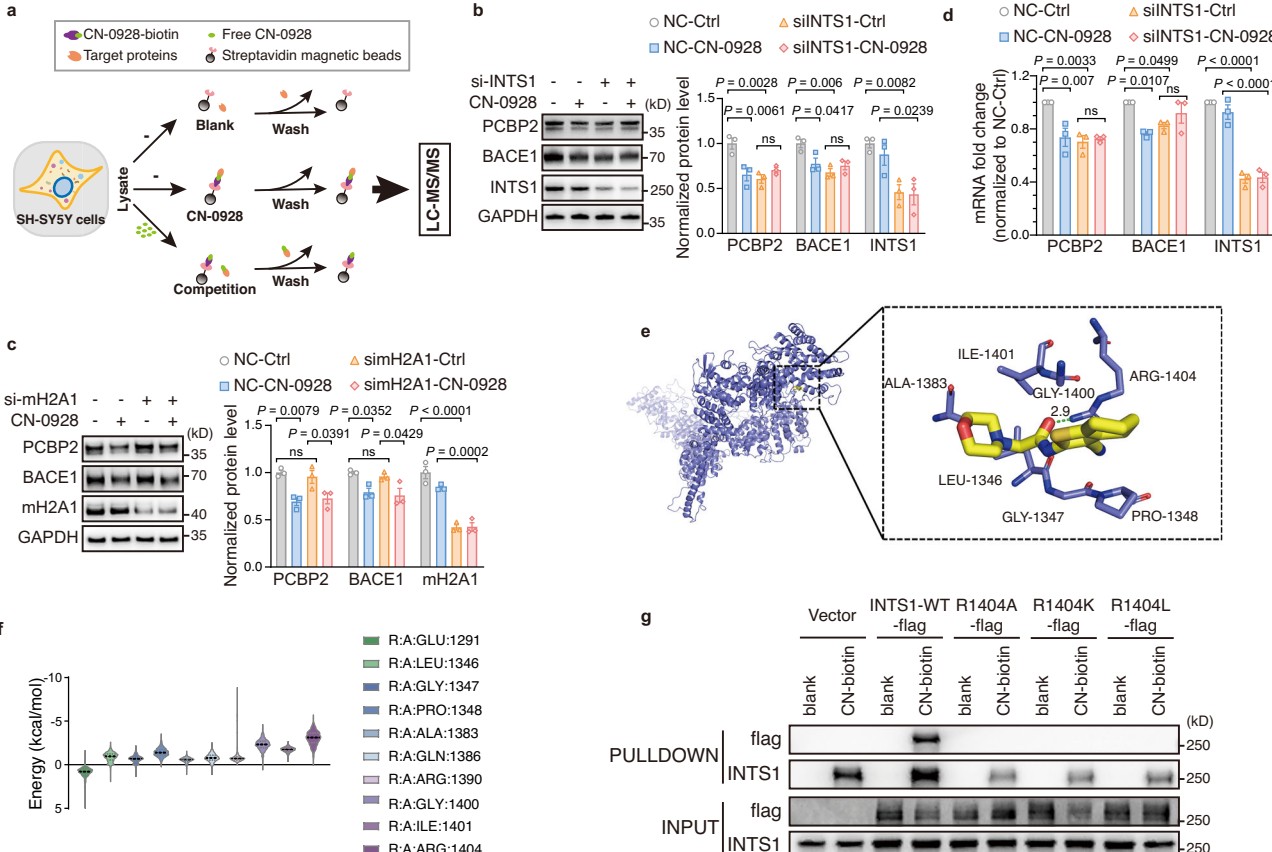

**Fig. 7 | CN-0928 downregulates PCBP2 protein by INTS1. a** The workflow of chemical proteomics for target identification of CN-0928 includes three groups: In the CN-0928 group (middle), SH-SY5Y cell lysate was incubated with CN-0928-conjugated beads to enrich putative CN-0928 target proteins. In the competition group (lower), the lysate was preincubated with excessive free CN-0928 and then incubated with CN-0928-conjugated beads. In the blank control (upper), beads without CN-0928 were used to identify nonspecifically bound proteins. Each experiment was performed in duplicate, and the proteins pulled down were analyzed by LC-MS/MS. **b, c** Representative Western blots (left) and quantification (right) of BACE1 and PCBP2 in SH-SY5Y cells with INTS1 (**b**) or mH2A1 (**c**) knock-down, followed by treatment with either 1 μM CN-0928 or DMSO (Ctrl). **d** Relative mRNA levels of PCBP2 and BACE1 in SH-SY5Y cells transiently transfected with siINTS1, measured in the absence (Ctrl) or presence of 1 μM CN-0928. **e** Molecular

docking results of CN-0928 to INTS1. The panoramic view shows the spatial position of CN-0928 in the structure of INTS1. The hydrogen bond is presented in the local interaction image. **f** Per-residue decomposition analysis of the binding free energy. In the CN-0928-INTS1 binary complex, CN-0928 forms hydrogen bonds with ARG-1404 during molecular dynamics simulations. Violin width is proportional to the kernel-density estimate of the distribution; the central line denotes the median and the thin lines indicate the 25th and 75th percentiles (*n* = 1000 analyzed frames). **g** CN-0928-biotin pulldown of flag-tagged wild type (INTS1-WT-flag) and ARG-1404 mutated INTS1 (R1404A-flag, R1404K-flag, and R1404L-flag) proteins. Cell lysates were mixed without and drug (blank), or with CN-0928-biotin (CN-biotin). Data: mean ± s.e.m (**b–d**). Significance was determined by one-way ANOVA with Bonferroni's test (**b–d**). ns nonsignificant. In vitro: *n* = 3 biological replicates (**b–d, g**). Source data are provided as a Source Data file.

It is reported that PCBP2 is a component of SG/PB[30,31]. However, whether PCBP2 undergoes LLPS is not well-established. In our study, a majority of PCBP2 condensates do not colocalize with Aβ fibrils (Supplementary Fig. 1j, k), suggesting a different identity from aggresomes. Moreover, PCBP2 condensates might not be identical to SG/PB,

because PCBP2 condensates recruit a large number of mitochondrial components, which, to the best of our knowledge, are not included in SG and PB[4,44]. A phase separation mechanism could be supported by the following: (1) PCBP2 condensates are chemically regulated by Ars and 1,6-HD that control intracellular protein condensation; (2) the

liquid-like droplet occurs when purified PCBP2 protein is incubated with RNA at physiological concentrations; and the fusion event develops along with time (Supplementary Fig. 2). (3) FRAP shows that PCBP2 droplet signals immediately vanished and quickly recovered within seconds in living cells; and (4) Selective deletion of the IDRs in the PCBP2 protein sequence prevents condensate formation (Fig. 5p, q), indicating a critical role of IDRs in phase separation. Our study also reveals a mitochondrial protein TOM20 that otherwise fails to independently develop visible clusters forms a droplet with PCBP2, supporting that TOM20 is a client protein of PCBP2. We speculate that the LLPS mechanism could be involved in the regulation of mitochondrial activity.

Mitochondrial dysfunction is believed to be fundamental to AD[49,65]. Indeed, several mitochondrial proteins tend to aggregate in AD brains and in cells challenged with Aβ[59,66]. In our study, many of the proteins recruited by PCBP2 condensates are critical for mitochondrial assembly[45,67], which may lead to a loss-of-function phenotype by reducing their cytosolic availability[3,68], correlating with the disruption of mitochondrial morphology in association with energy production and ROS homeostasis. It is tempting to speculate that an aberrant spatial clustering of proteins is involved in the regulation of mitochondrial activity by PCBP2 condensates. Deregulated protein compartmentalization offers a layer of disease mechanism beyond the altered expression levels of mitochondrial mRNA and protein in AD by genomics and proteomics studies[69,70].

How biomolecular condensates may regulate amyloidogenesis is still poorly understood. Aβ is mainly generated in the membranous endosome or multivesicular body before being exocytosed into the extracellular space[16]; therefore, it is unlikely that phase separation directly regulates amyloid deposition. In our study, PCBP2 condensates suppress the decay of BACE1 mRNA via a finely compartmentalized intracellular mechanism, thereby indirectly modulating Aβ deposition. In line with the role of biomolecular condensates in RNA metabolism[2], an involvement of NMD mechanism in BACE1 mRNA decay that is regulated by PCBP2 condensates is supported by the following findings: (1) PCBP2 sequesters regulatory proteins in NMD including UPF1, and silencing of UPF1 prevents the effect of PCBP2 on BACE1 mRNA level; (2) Outside the condensates, the spatial abundance of 3′UTR clusters is inversely correlated with that of UPF1, indicating that PCBP2 condensates sequester UPF1 and thereby protect extra-condensate 3′UTRs from efficient degradation; (3) Condensation-deficient PCBP2 mutants have no regulatory effect on BACE1. Overall, these data agree with the hypothesis that Aβ deposition, which results from an altered BACE1 translation, may be a by-product of biomolecular condensates[1], highlighting a different mechanism relative to tau/TDP-43 aggregation. On the other hand, the same mechanism of protein sequestration may be involved in the regulation of both amyloid deposition and mitochondrial dysfunction by PCBP2 condensates, thus integrating the seemingly different but important pathways in the pathophysiology of AD.

Protein aggregation is a hallmark of AD; thus, targeting biomolecular condensates may be particularly important. In our study, PCBP2 condensation correlates with an elevated protein level in cells and the brain of mice, which provides a rationale that reducing PCBP2 protein would inhibit condensation. We identify that CN-0928 binding to the target protein INTS1 allows control of PCBP2 expression, possibly through a transcriptional mechanism, and the reduced PCBP2 protein level and condensation are then responsible for the alleviated AD pathologies. Integrator complex proteins regulate transcription, DNA repair, and RNA binding[71], whereas a connection between INTS1 and PCBP2 suggests that INTS1 could also be involved in neuronal senescence and neurodegenerative diseases through biomolecular condensates[72]. We show that CN-0928 reduces PCBP2 condensates, relieves mitochondrial damage and reduces BACE1 and Aβ generation, contributing to an improvement of cognitive function in AD mice.

These data provide proof-of-concept that therapies aimed at reducing biomolecular condensates may offer an effective strategy for treating condensate-associated disorders, including AD[10,73].

This study also has several limitations. Target identification with CN-0928 was carried out in cell lysates (overnight at 4 °C), while phenotypic screening was performed in live cells (48 h at 37 °C) and in mouse models (one month at 37 °C). This discrepancy introduces differences in CN-0928 behavior between live cells and lysates under varying conditions, which could lead to variability due to the absence of certain protein-protein and protein-nucleic acid interactions in lysates. Moreover, CN-0928-biotin does not alter PCBP2 protein level in cultured cells, possibly due to the poor membrane permeability, which prevents further testing of CN-0928-biotin in animal models. Nonetheless, the application of chemical biology facilitates the discovery of INTS1 as a target protein of CN-0928, highlighting a nuclear mechanism involved in the pathology of AD, which may deserve further investigation.

## Methods

### Antibodies

Antibodies against the following proteins or epitopes were purchased from the indicated sources: ADAM10 (pAb, Abcam, catalog no. ab1997, 1:1000); APP, C-Terminal (pAb, Sigma, catalog no. A8717, 1:1000); β-Amyloid, 1-16 [6E10] (mAb, BioLegend, catalog no. 803014, 1:500); BACE1 (pAb, Abcam, catalog no. ab2077, 1:1000); BACE1 [EPR3956] (mAb, Abcam, catalog no. ab108394, 1:2000); BACE1 [EPR19523] (mAb, Abcam, catalog no. ab183612, 1:200); PCBP2 [EPR14858] (mAb, Abcam, catalog no. ab184962, 1:1000 for WB; 1:200 for IF); RENT1/hUPF1 [EPR4681] (mAb, Abcam, catalog no. ab109363, 1:1000); phospho-Upf1 (Ser1127) (pAb, Sigma, catalog no. 07-1016, 1:1000); TIA1 (mAb, Santa Cruz, catalog no. sc-166247, 1:50); DDX6 (mAb, Santa Cruz, catalog no. sc-376433, 1:50); DCP1a (mAb, Santa Cruz, catalog no. sc-100706, 1:50); TOMM20 [EPR15581-39] (mAb, Abcam, catalog no. ab283317, 1:1000 for WB; 1:200 for IF); NeuN [1B7] (mAb, Abcam, catalog no. ab104224, 1:200); Iba1 [EPR16589] (mAb, Abcam, catalog no. ab283346, 1:200); GFAP [EPR1034Y] (mAb, Abcam, catalog no. ab279291, 1:200); INTS1(pAb, Proteintech, catalog no. 31428-1-AP, 1:1000); Histone Macro-H2A.1 (mAb, Invitrogen, catalog no. MA5-24696, 1:1000); TUBGCP2 (pAb, Proteintech, catalog no. 25856-1-AP, 1:1000); HSP40 (mAb, Santa Cruz, catalog no. sc-398766, 1:50); DNAJC16 (pAb, FineTest, catalog no. FNab02461, 1:1000); Flag [DYKDDDDK tag] (mAb, Proteintech, catalog no. 66008-4-Ig, 1:1000); GFP (mAb, Proteintech, catalog no. 66002-1-Ig, 1:1000); mCherry (mAb, Proteintech, catalog no. 68088-1-Ig, 1:2000); GAPDH (mAb, Proteintech, catalog no. 60004-1-Ig, 1:10,000); β-Actin (mAb, Proteintech, catalog no. 66009-1-Ig, 1:10,000); HRP-conjugated Affinipure Goat Anti-Mouse IgG(H+L) (Proteintech, catalog no. SA00001-1, 1:10,000); HRP-conjugated Affinipure Goat Anti-Rabbit IgG(H+L) (Proteintech, catalog no. SA00001-2, 1:10,000); Goat Anti-Rat IgG H&L (Alexa Fluor® 647) preadsorbed (Abcam, catalog no. ab150167, 1:250); Goat Anti-Rabbit IgG H&L (Alexa Fluor® 568) (Abcam, catalog no. ab175471, 1:250); Goat Anti-Mouse IgG H&L (Alexa Fluor® 488) (Abcam, catalog no. a b150113, 1:250).

### Reagents

The following reagents were purchased from Thermo Fisher Scientific: FastDigest NheI (catalog no. FD0974), FastDigest BamHI (catalog no. FD0054), TRIzol™ (catalog no. 15596026CN), Dulbecco's Modified Eagle Medium (DMEM) (catalog no. 11965092), Dulbecco's Modified Eagle Medium/Nutrient Mixture F-12 (DMEM/F-12) (catalog no. 11320033), Opti-MEM™ I (catalog no. 31985070), Fetal Bovine Serum (FBS) (catalog no. 10099141C), Lipofectamine™ 3000 (catalog no. L3000015), Lipofectamine™ RNAiMAX (catalog no. 13778150). The following reagents were purchased from MedChemExpress: Actinomycin D (catalog no. HY-17559), DAPT (catalog no. HY-13027), Puromycin (catalog no. HY-K1057), Protease Inhibitor Cocktail (EDTA-

Free) (catalog no. HY-K0010), Streptavidin Magnetic Beads (catalog no. HY-K0208), and Thioflavine S (catalog no. HY-D0972). The following reagents were purchased from Beyotime: Hoechst (catalog no. C1028), Penicillin-Streptomycin Solution (catalog no. C0222), G418 Sulfate (catalog no. ST081), Reactive Oxygen Species Assay Kit (catalog no. S0033S), MitoTracker Green (catalog no. C1048), and Dual-Lumi™ II Luciferase Assay Kit (catalog no. RG089S). The following reagents were purchased from Vazyme: ClonExpress Ultra One Step Cloning Kit (catalog no. C115-01), 2× Phanta Max Master Mix (Dye Plus) (catalog no. P525-01), HiScript II Q Select RT SuperMix for qPCR (+gDNA wiper) (catalog no. R233-01), ChamQ Universal SYBR qPCR Master Mix (catalog no. Q711-02). The following reagents were purchased from Promega: RNasin® Ribonuclease Inhibitor (catalog no. N2115), RQ1 RNase-Free DNase (catalog no. M6101), and Riboprobe® In Vitro Transcription Systems (catalog no. P1440). The recombinant human PCBP2 protein and TOM20 protein (catalog no. P1247) were purchased from GENE CREATE and Finetest, respectively. The following reagents were purchased from Abcam: DyLight® 488 Conjugation Kit (catalog no. ab201799) and Alexa Fluor® 555 Conjugation Kit (catalog no. ab269820). CN-0928 (CID 826287), CN-0928-HCL, and CN-0928-biotin were synthesized and purified in Nie's lab.

### Plasmids
All plasmids used in this study are described in Supplementary Data 6.

For expression in mammalian cells, cDNAs for PCBP2-WT, PCBP2-del IDR1, PCBP2-del IDR2, PCBP2-del IDR3, PCBP2-del IDR4, INTS1-WT, INTS1mut1 (R1404A), INTS1mut2 (R1404L), and INTS1mut3 (R1404K) were synthesized and cloned into pcDNA3.1(+), in which each protein was fused with a 3×FLAG tag at the C terminus. pcDNA3.1-MS2-BACE1 3′ UTR/5′UTR, pMCP-GFP, and pmirGLO-BACE1 3′UTR/5′UTR were purchased from YouBio (Changsha, China). VN-dEcCas6-VC-GK construct and Actin-8×/16×CBS-GK construct were kindly gifted by Dr. Chun-Yu Han (Hebei University of Science and Technology)[74]. The target fragments (BACE1 3′UTR) were inserted into the 16×CBS-GK construct (double digested and amplified from the Actin-16×CBS construct) to generate the BACE1 3′UTR-16×CBS-GK construct. All recombinant plasmids were verified by DNA sequencing (Sangon Biotech, Shanghai, China).

### siRNA and DNA transfection
The small interfering RNA (siRNA) and scrambled siRNA as a negative control (NC) were designed and synthesized by Sangon Biotech. siRNA sequences are listed in Supplementary Data 6.

siRNA and DNA transfections were performed using Lipofectamine RNAiMAX or Lipofectamine 3000, respectively, according to the manufacturer's instructions. Western blotting or RT-qPCR was conducted 48–72 h post-transfection to evaluate transfection efficiency.

### Cell culture
SH-SY5Y cells (National Collection of Authenticated Cell Cultures, catalog no. SCSP-5014) were maintained in a 1:1 mixture of DMEM-F12, and HEK293T (ATCC, catalog no. CRL-3216) cell lines were cultured in DMEM, supplemented with 10% FBS, 100 U/mL penicillin G, and 100 mg/mL streptomycin sulfate. The SH-SY5Y cell line most frequently used in this paper was validated by STR profiling, showing an exact match with the SH-SY5Y line in the ExPASy STR database based on 13 core STR loci. Other cell lines used in this study were not authenticated. HEK293T-APP and SH-SY5Y-APP cell lines that produce high Aβ levels were used as a cellular model of AD. These cell lines stably expressing human wild-type amyloid-beta precursor protein (APP) 695 were maintained in the same media with an additional 200 µg/mL G418[75]. For the stable expression of mCherry-PCBP2, SH-SY5Y and SH-SY5Y-APP cells were transduced with LV6-mCherry-PCBP2 and LV6-mCherry vectors. Forty-eight hours post-transduction, the cells were transferred to 10 cm dishes and supplemented with 2 µg/mL puromycin. Clones expressing mCherry-PCBP2 were identified and selected using fluorescence microscopy.

### Mouse models
APP/PS1 mice (APPswe, PSEN1dE9, B6C3, #034829-JAX) and 5×FAD mice (APPSwFlLon, PSEN1*M146L*L286V, B6SJL, #034840-JAX) transgenic mice were purchased from Jackson Laboratory or GENEANDPEACE, respectively. Mice were maintained in a specific-pathogen-free facility in individually ventilated cages (3–5 per cage) on a 12-h light/12-h dark cycle, at an ambient temperature of $22 \pm 2\,°C$ and relative humidity of 40–60%. Standard chow and water were provided ad libitum, with nesting materials and environmental enrichment. Animals were acclimated for at least 7 days before experimentation. Wild-type (WT) mice of the same genetic background were identified by genotyping; only adult male mice were included and randomly assigned to experimental groups. Mice with exact $n$ and ages reported in the figure legends were used in this study.

To evaluate the effects of CN-0928 and CN-0928-HCl, 5-month-old 5×FAD mice were intraperitoneally injected with CN-0928 (3.5 mg/kg), CN-0928-HCl (3.5 mg/kg), or saline every other day for one month. All mice were anesthetized with pentobarbital (50 mg/kg, i.p.) and perfused with chilled saline. One hemisphere was fixed in 4% paraformaldehyde (PFA) and then embedded in paraffin for immunofluorescence. The other hemisphere, including the hippocampus and cerebral cortex, was rapidly isolated on ice and immediately snap-frozen in liquid nitrogen for biochemical analysis.

### Human brain tissues
Postmortem human brains utilized in this study were sourced from the Human Brain Bank at Xiangya School of Medicine, obtained through its voluntary donation program. Clinical information of patients is presented in Supplementary Table 1. All samples were de-identified; only exact age and sex were retained as indirect identifiers, which are insufficient to identify any individual participant. All tissue was either consumed during experiments or immediately and irreversibly destroyed after analysis per the bank's SOPs; no residual material was retained.

### FAPS of PCBP2-biomolecular condensates
For each sorting experiment, cells were cultured to approximately 80–90% confluence in three 15-cm dishes. Cells were pelleted in PBS by centrifugation at $1000 \times g$ for 3 min at RT and immediately flash-frozen in liquid nitrogen overnight. Unless otherwise stated, the following steps were conducted at 4 °C or on ice. Cell pellets were resuspended in lysis buffer (50 mM Tris [pH 7.4], 1 mM EDTA, 150 mM NaCl, 0.2% Triton X-100), containing 65 U/mL RNase inhibitor and EDTA-free protease inhibitor cocktail. To facilitate lysis, the extracts were passed 20 times through a 25G syringe needle, with a total incubation time of 20 min. Lysates were spun at $200 \times g$ for 5 min to remove nuclei. To digest remaining DNA contaminants, the supernatants were treated with RQ1 RNase-free DNase for 30 min at RT, and then centrifuged at $10,000 \times g$ for 7 min. The resulting pellets were resuspended in 2 mL of lysis buffer, and the organelle-enriched fraction obtained at $10,000 \times g$ was designated as the pre-sorted fraction. From this fraction, PCBP2-biomolecular condensates were sorted on a cell sorter (Bigfoot, Invitrogen) equipped with a 100 µm nozzle and operating at a pressure of 30 psi. Particles were identified based on their forward-scattered light (FSC) and mCherry fluorescence, utilizing a 561 nm excitation laser and a 615/24 nm bandpass filter. The sorting window for PCBP2-biomolecular condensates was specifically set to exclude mCherry-SH-SY5Y fluorescent particles while retaining mCherry-labeled PCBP2 condensates. The mCherry-PCBP2 fraction represented 8-10% of total events and was subsequently collected. These fractions were centrifuged at $10,000 \times g$ for 7 min, and the pellets were then analyzed as the post-sorted fractions.

## Microproteomics of PCBP2-biomolecular condensates

The pre-sorted and post-sorted fractions were commissioned to Beijing Genomics Institute (BGI). This project used the high-resolution mass spectrometer Orbitrap Fusion Lumos Tribrid (Thermo Fisher Scientific, San Jose, CA) to perform data-dependent acquisition (DDA) mode detection on library samples and micro-samples. The integrated MaxQuant's Andromeda engine was used for identification analysis of the library samples and micro-samples. The identification results of the micro-sample data were filtered at the spectrum level with PSM-level FDR <= 1%, and then further filtered at the protein level with Protein-Level FDR <= 1% to obtain significant identification results.

## Western blotting

Cells or brain tissues were homogenized in ice-cold lysis buffer (Beyotime Biotechnology), with complete protease inhibitor cocktail (Med Chem Express). Samples were sonicated on ice for 30 min and centrifuged at $16,000 \times g$ for 15 min, and the resultant supernatant was collected. Denatured protein samples were separated on an 8% SDS-PAGE gel, and a 16.5% Tris-tricine gel was used specifically for the detection of CTFs in the presence of γ-secretase inhibitor DAPT (250 nM) for 24 h. Blot images were visualized using the Fusion FX5 image analysis system (Vilber Lourmat).

## RNA isolation and real-time quantitative PCR (RT-qPCR)

Total RNA was extracted using TRIzol and the RNA Easy Fast kit. cDNA synthesis was performed using the Reverse Transcriptase Kit. mRNA levels were quantified using the SYBR Green PCR Master Mix and the Bio-Rad CFX96 Real-Time PCR Detection System. Primer sequences are specified as in Supplementary Data 6. Each experiment included three biological replicates and at least three technical replicates. Relative mRNA levels were quantified using the ΔΔCt method and expressed as fold changes. The RIP-qPCR data were analyzed following previously described methods[76].

## Live-cell imaging and immunofluorescence

Immunofluorescence was performed as previously described[75]. Briefly, for live-cell imaging of mitochondria and reactive oxygen species (ROS), $5 \times 10^5$ SH-SY5Y, SH-SY5Y-APP, or SH-SY5Y-mCherry-PCBP2 cells were seeded in 35 mm confocal dishes 12–18 h and then treated with 50 μM hydrogen peroxide for 4 h as a positive control. Cells were stained with 20 nM MitoTracker Green or 10 μM DCFH-DA for 20–30 min at 37 °C.

For RNA imaging by fluorescence-enrichment platform, $1 \times 10^6$ SH-SY5Y were seeded in 35 mm confocal dishes 12–18 h before transfection, and then these cells were co-transfected with 1.5 μg pcDNA3.1-MS2-BACE1 3′UTR and 0.5 μg pMCP-GFP per well for 24–48 h. Cells were then stressed by the addition of sodium arsenite (0.5 mM) for 30 min. After that, cells were fixed with 4% PFA for 15 min at 37 °C followed by permeabilization with 0.3% Triton X-100 for 15 min. After being stained with DAPI and PCBP2, the cells were observed with a Leica confocal microscope (TCS SP8 X).

For RNA imaging by fluorescence-activation platform, $1 \times 10^6$ SH-SY5Y or SH-SY5Y-mCherry-PCBP2 cells were seeded in 35 mm confocal dishes 12–18 h before transfection, and then these cells were co-transfected with 0.8 μg VN-dEcCas6-VC-GK and 3.2 μg BACE1 3′UTR-16×CBS-GK. After 24–48 h, cells were imaged in a heated chamber (37 °C, 5% CO2) on a Leica THUNDER Imager Live Cell.

For the observation of PCBP2 condensates in live cells, $5 \times 10^5$ SH-SY5Y-mCherry-PCBP2 cells were seeded in 35 mm confocal dishes and cultured for 12–18 h. Subsequently, the cells were treated with 1.5% 1,6-hexanediol (1,6-HD) and immediately subjected to time-lapse imaging using a confocal microscope. Images were acquired every 1 min for the first 15 min, during which larger PCBP2 condensates visibly dissolved. To minimize photobleaching, acquisition was then reduced to one frame every 2 min and continued for 60 min.

Thioflavin S staining was performed after completion of the PCBP2 primary and secondary antibody labeling steps. Samples were rinsed three times in PBS (5 min each), then incubated in the dark for 8 min at room temperature with Thioflavin S (0.05% w/v for tissue sections or 0.1% w/v for cultured cells, prepared in distilled water). Immediately thereafter, slides were washed twice in 80% ethanol (1 min each) and twice in PBS (5 min each). Specimens were mounted in antifade medium lacking DAPI and imaged with a Leica confocal microscope (TCS SP8 X).

## Phase-separation

For fluorescence observation of phase separation, full-length recombinant human TIA1 and PCBP2 proteins were fluorescently labeled using the DyLight® 488 Conjugation Kit and Alexa Fluor® 555 Conjugation Kit, following the manufacturers' instructions. To achieve optimal quantum yield for the experimental reaction, the fluorescently labeled proteins were mixed with unlabeled proteins from the same purification batch at a ratio of 1:4–1:5. Unless otherwise specified, the phase separation observations were conducted in the following system: 5 μM recombinant human protein, 10 mM HEPES (pH 7.4), 100 mM NaCl, 10% PEG-8000, and 1.5% glycerol. After incubating the above mixture at room temperature for 2 h, differential interference contrast (DIC) and fluorescence imaging were performed using a Leica confocal microscope (TCS SP8 X).

## Fluorescent recovery after photobleaching (FRAP)

FRAP experiments were conducted on liquid droplets formed by PCBP2-A555 induced by 10% PEG-8000, using a Leica confocal microscope equipped with a 63× objective lens. SH-SY5Y cells were seeded into confocal dishes and transfected the following day. Live-cell FRAP was performed 48 h post-transfection on a Leica confocal microscope (TCS SP8 X) using a 63× objective. SH-SY5Y-mCherry-PCBP2 cells were seeded into confocal dishes and subjected to FRAP the following day on the same Leica system with a 63× objective. FRAP imaging was performed using the FRAP module of the Leica confocal microscope in ROI mode. Two pre-bleaching images were captured as a baseline, followed by bleaching of the selected ROI. Post-bleaching images were acquired at 2-s intervals over a total duration of 2 min. The bleached ROI was analyzed using the FRAP Profiler plugin in ImageJ, with unbleached regions serving for background normalization and drift correction. Curve fitting was subsequently performed using GraphPad Prism.

## Transmission electron microscopy (TEM)

Cells were washed twice with pre-warmed PBS, pre-fixed with 2.5% glutaraldehyde solution at RT for 20 min, scraped, and centrifuged at $1000 \times g$ for 5 min. Cell pellets were gently fixed with 2.5% glutaraldehyde solution at 4 °C overnight. Then, samples were washed three times in 0.1 M phosphate buffer (pH 7.0), post-fixed with 1% osmium tetroxide solution for 1–2 h, followed by three washes in 0.1 M phosphate buffer and subsequent dehydration in alcohol (15 min each; 30%, 50%, 70%, 80%, 90%, and 100%) and acetone for 20 min. Samples were then placed in a mixture of acetone and resin 3:1 (2 h), 1:1 (3 h), 1:3 (3 h), and finally in resin overnight. Samples were infiltrated and embedded in a mold, and before gradient heating (35 °C–60 °C–80 °C, 5 h in each). Sections of 70 nm were cut on LEICA EM UC7 ultramicrotome and imaged using a JEM-1200 EX electron microscope with an accelerating voltage of 120 kV. Mitochondrial numbers were counted manually, and mitochondrial morphology was measured by ImageJ software.

## Enzyme-Linked Immunosorbent Assay (ELISA)

Concentrations of solutes in culture medium and cell lysates were quantified using an ELISA Kit following the manufacturer's protocol. Both soluble and formic acid-insoluble fractions of Aβ1-42

(Elabscience, catalog no. E-EL-H0543) and Aβ1-40 (Elabscience, catalog no. E-EL-H0542) were isolated in accordance with previously published methods[77].

## Luciferase activity assay

Cells were seeded in a black 96-well plate with 5000 cells per well and transfected with the luciferase reporter plasmids for 48 h. The luciferase activity was measured by a GloMax 96 microplate luminometer (Promega) using the Dual-Lumi II Luciferase Reporter Gene Assay Kit.

## Metabolic assay

The oxygen consumption rate (OCR) and extracellular acidification rate (ECAR) were assessed using a Seahorse XF24 Extracellular Flux Analyzer (Agilent Technologies, Santa Clara, CA, USA) with the XF Cell Mito Stress Test Kit (Agilent, catalog no. 103015-100) and the XF Glycolysis Stress Test Kit (Agilent, catalog no. 103020-100), respectively. SH-SY5Y and SH-SY5Y-PCBP2 cells were cultured in an XF24 cell culture microplate. To evaluate mitochondrial respiration, including basal and maximal respiration, proton leak, and ATP production, cells were treated with oligomycin, FCCP, and a combination of rotenone and antimycin A. Additionally, glucose, oligomycin, and 2-deoxy-glucose (2-DG) were introduced to perform glycolysis stress tests.

## RNA pulldown combined with LC-MS/MS

The RNA pulldown assay was conducted using the RiboTrap Kit (MBL, catalog no. RN1012). Briefly, 5-bromo-UTP (BrU) was randomly incorporated into the 3′ UTR of BACE1 during in vitro transcription with the Riboprobe in vitro Transcription Systems-T7 (Promega). The transcription utilized a linearized pcDAN3.1(+)-BACE1 3′UTR as the template. Subsequently, cell lysates (cytoplasmic extract without nuclei) were incubated with the resultant anti-BrdU conjugated beads at 4 °C for 2 h. After washing and eluting the samples, they were either sent for peptide analysis via LC-MS/MS at Applied Protein Technology (Shanghai, China) or assessed through immunoblotting for verification.

## MS2-Trap

Cells were cultured in 15-cm dishes to approximately 60–70% confluence and co-transfected with pcDNA3.1-MS2-BACE1 3′UTRΔ/5′UTR/3′UTR and pMCP-GFP as previously described. After 48 h, confluence reached approximately 80–90%. MS2-GFP substrate conjugates were then isolated using the GFP-Trap® Magnetic Agarose Kit (ChromoTek, catalog no. gtmak), following the manufacturer's instructions. The isolated pellets were subsequently mixed with SDS-loading buffer in preparation for Western blot analysis.

## RNA Immunoprecipitation (RIP)

RIP assays were conducted using the Magna RIP RNA-binding protein immunoprecipitation kit (Sigma-Aldrich, catalog no. 17-701) following the manufacturer's instructions. SH-SY5Y cells were cultured in 15-cm dishes to 60–70% confluence and transfected with various plasmids (Vec/FL-PCBP2/KH1-PCBP2/KH2-PCBP2/KH3-PCBP2). After 48 h, cells of 80–90% confluence were harvested using RIP lysis buffer and then incubated at 4 °C overnight with gentle rotation, using either 6 μg of DYKDDDDK tag antibody or a corresponding volume of control IgG that was conjugated to magnetic beads. Following stringent washing, RNAs were preserved at −80 °C and analyzed via RT-qPCR assays.

## Chemical proteomics and target validation

SH-SY5Y cells cultured in 15-cm dishes were lysed using an ice-cold lysis buffer containing 20 mM Tris (pH 7.5), 150 mM NaCl, 0.1% NP-40, and 1 mM EDTA, along with a protease inhibitor cocktail. The cell extracts were initially incubated with or without CN-0928 (100 μM) at 4 °C overnight. Following this, either CN-0928-biotin-immobilized streptavidin magnetic beads or CN-0928-free beads were added to each sample and incubated at 4 °C for an additional 4 h. The beads were then washed five

times, and the resulting pulldown pellets were combined with SDS-loading buffer and subsequently analyzed by LC-MS/MS and Western blot. For the target validation experiment, SH-SY5Y cells were seeded into 15-cm culture dishes. When cell confluency reached 60–70%, the plasmids pcDNA3.1(+)-3×flag-INTS1, pcDNA3.1(+)-3×flag-INTS1mut1 (R1404A), or pcDNA3.1(+)-3×flag-INTS1mut2 (R1404L) were transfected into the cells. After 48–72 h of transfection, the cells were lysed on ice. The pulldown results were analyzed by Western blot.

For ligandability analysis, scatter plots were generated by applying a $\log_{10}$ transformation to the iBAQ mean values of proteins in the CN-0928 group and the Competition group, excluding those in the Blank group. Proteins that were detected in the CN-0928 group but were not detected in either the Competition group or the Blank group were selected as candidate target proteins.

## Memory performance testing

Spatial memory was tested by Morris water maze[78]. This procedure comprised a one-day visible platform test (four consecutive trials with a 60-min intertrial interval) and a five-day training phase (four trials per day), culminating in a probe trial 24 h after the final training session. During this probe trial, mice were placed on the opposite side of the original platform quadrant, and metrics such as the number of entries and duration within the original quadrant were recorded. Additionally, the escape latency-the time taken by each mouse to locate the hidden platform-was noted. All animal trajectories were tracked and analyzed using the computerized software SuperMaze (Shanghai Xinran Corporation).

### Synthesis of 2-chloro-N-(3-cyano-4,5,6,7-tetrahydrobenzo[b]thiophen-2-yl) acetamide (Supplementary Fig. 8a)

To a solution of 2-amino-4,5,6,7-tetrahydrobenzo[*b*]thiophene-3-carbonitrile (**1**, 5.0 g, 28.0 mmol, 1 eq.) In 1,4-dioxane (20.0 mL) was added chloroacetyl chloride (**2**, 6.3 g, 56.0 mmol, 2eq.) drop-wise over a period of 10 min by dropping funnel at 0 °C. The reaction was allowed to warm to room temperature and stirred for another 20 min. The reaction was monitored by TLC. Upon completion of the reaction, the mixture was poured into ice-cold water. The desired product was collected by filtration, and further triturated using a mixture of diethyl ether: hexane (80:20,v/v) to give **2** (6.2 g, 87%) as an off-white solid.

## General procedure for synthesis of (5a-b) (Supplementary Fig. 8a)

A mixture of intermediate **3** (0.78 mmol, 1 eq.) and triethylamine (2.35 mmol, 3 eq.) in 1,4-dioxane (3 mL) was stirred at room temperature for 10 min, followed by the addition of **4a-b** (0.78 mmol, 2 eq.). Then the reaction was heated to 70 °C and stirred until the completion of the reaction, cooled to room temperature, and the reaction mixture was poured into ice-cold water. Precipitates were obtained and collected using vacuum filtration and dried. The crude products were purified by trituration using diethyl ether to give **5a-b** as an off-white solid.

tert-butyl 4-(2-((3-cyano-4,5,6,7-tetrahydrobenzo[b]thiophen-2-yl)amino)-2-oxoethyl)piperazine-1-carboxylate (**5a**)

Compound **5a** was synthesized according to the general procedure described above as an off-white solid in a yield of 63%.

N-(3-cyano-4,5,6,7-tetrahydrobenzo[b]thiophen-2-yl)-2-morpholinoacetamide (**5b**)

Compound **5b** was synthesized according to the general procedure described above as an off-white solid in a yield of 50%.

### Synthesis of N-(3-cyano-4,5,6,7-tetrahydrobenzo[b]thiophen-2-yl)-2-(piperazin-1-yl) acetamide (CN-0928-HCl)

To a solution of intermediate **5a** (150 mg, 0.37 mmol, 1eq) in dry dichloromethane (5.0 mL) was added HCl (4 M in 1,4-dioxane, 0.46 mL). The resulting reaction mixture was stirred at room temperature and stirred for 3 h. The reaction was monitored by TLC. Upon completion of the reaction, the mixture was concentrated under a vacuum to yield crude material as an off-white powder.

## Synthesis of N-(3-cyano-4,5,6,7-tetrahydrobenzo[b]thiophen-2-yl)-2-(4-(5-((3aS,4R,6aR)-2-oxohexahydro-1H-thieno[3,4-d]imidazol-4-yl)pentanoyl)piperazin-1-yl)acetamide (CN-0928-biotin)

To a solution of N-(3-cyano-4,5,6,7-tetrahydrobenzo[b]thiophen-2-yl)-2-(piperazin-1-yl)acetamide (**6a**, 200 mg, 0.65 mmol, 1 eq.) in dry dichloromethane (10.0 mL) was added HATU (75 mg, 1.97 mmol, 3 eq.). The resulting mixture was stirred at room temperature for 1 h, then DIPEA (59 mg, 4.6 mmol, 7 eq) and D-Biotin (321 mg, 1.30 mmol, 2 eq.) were added and stirred at room temperature overnight. After the reaction was completed, the solvent was evaporated under reduced pressure, then $NaHCO_3$ saturated solution was added and extracted with EA (×3). Crude **CN-0928-biotin** was purified by column chromatography using petroleum ether: ethyl acetate (5:1) as a mobile phase to afford a white solid (98 mg, 28%).

### Quantification, graphing, and statistical analysis

All images were analyzed and quantified using ImageJ. Data were plotted, and statistical analysis was performed using Graphpad Prism 9.5.1 software. Fluorescence images were acquired using LAS X (Leica), with the LIGHTNING module employed for super-resolution imaging and the THUNDER module for defocus elimination. Flow cytometry was collected and analyzed with SQ Software (Invitrogen) and FlowJo version 10.8.1, respectively. Metabolic analysis data were acquired and processed using Wave software version 2.6.1.53 (Agilent). Molecular docking experiments were conducted using MOE2019, and Gromacs2018 was used as the molecular dynamics simulation software.

### Ethical statement

All human and animal research complied with relevant ethical guidelines and was approved by the appropriate committees. The human study was approved by the Institutional Review Board of the School of Basic Medical Science, Central South University (approval no. 2020KT-37). Procedures conformed to the Ethical Review of Biomedical Research Involving Human Subjects, ICH-GCP, and the principles of the Declaration of Helsinki, and followed the Standardized Operational Protocol established by the China Human Brain Banking. Written informed consent was obtained from all participants. All animal experiments were approved by the Institutional Animal Care and Use Committee of Chongqing Medical University (approval no. IACUC-CQMU-2024-0023) and conducted in accordance with international standards.

### Reporting summary

Further information on research design is available in the Nature Portfolio Reporting Summary linked to this article.

## Data availability

Unless otherwise stated, all data supporting the results of this study can be found in the article, supplementary, and source data files. RNA sequencing data of PCBP2 knockdown in SH-SY5Y cells have been deposited in the Gene Expression Omnibus (GEO) under the accession numbers GSE306042. The mass spectrometry proteomics data have been deposited to the ProteomeXchange Consortium via the iProX partner repository with the dataset identifier PXD055459 [https://www.iprox.cn/page/project.html?id=IPX0009568000]. Raw spectral characterization data for all synthesized small molecules have been deposited in Figshare and are publicly available at [https://doi.org/10.6084/m9.figshare.29972815]. Source data are provided with this paper.

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

## Acknowledgements

We thank L.-Q. Ma for technical assistance, and C.-Y. Han at Hebei University of Science and Technology for providing VN-dEcCas6-VC and actin-8×CBS plasmids. This work was supported by National Natural Science Foundation of China (NSFC grants 82571619 and 82271461 to G.-J.C.), Deng-Feng Plan of Chongqing Medical University (grant cyyy-xkdfjh-jcyj-202304 to G.-J.C.), and Chongqing Ying-Cai Plan (grant CQYC20210309560 to G.-J.C.), and NSFC (Grant No. 82201578 to X.-J.X.).

## Author contributions

L.W. performed cell-free and cell culture experiments and analyzed human and mouse brain tissues. Q.-L.P. performed small-molecule screening and cell culture experiments. J.W.Z. performed chemical synthesis and biotinylation of CN-0928. G.-F.Z. conducted 3'UTR-targeting RBP analysis and data acquisition. S.Y.N. designed and supervised experiments on CN-0928 synthesis and labeling. Q.-L.Z. and X.-X.Y. provided human postmortem brain tissues. Y.X. analyzed proteomics and transcriptome data. L.W., X.-Y.X., and C.-L.L. performed animal experiments. Y.H. and X.-J.D. assisted with cell-free and cell culture experiments. X.-J.X., Y.-J.W., and J.-Y.Z. contributed to experimental design. G.-J.C. designed the experiments, conceived and supervised the study, and acquired funding. L.W. and G.-J.C. prepared the manuscript.

## Competing interests

G.-J.C. is the inventor on a patent application(s) owned by the First Affiliated Hospital of Chongqing Medical University related to CN-0928. The remaining authors declare no competing financial interests.
