## [Transparent Peer Review file · Nature Communications]

Pharmacologic inhibition of PCBP2 biomolecular condensates relieves Alzheimer's disease

Corresponding Author: Dr Guojun Chen

Version 0:

Reviewer comments:

Reviewer #1

(Remarks to the Author)

The revised manuscript by Wang et al. provides a thorough and insightful exploration of a specific pathological pathway in Alzheimer's disease and proposes potential therapeutic interventions. The authors have made a commendable effort to address the concerns raised in the previous review round. Below are a few minor comments on the revised submission. Upon satisfactory responses, I recommend the acceptance of this article.

Comments:

1. In Fig. 1b, the authors performed immunostaining of temporal cortex tissue sections from AD patients and non-AD individuals, where PCBP2 (red pseudocolor) indicates protein clustering. The authors were asked to clarify whether these clusters represent biomolecular condensates or amorphous protein aggregates, potentially sequestered in aggresomes or other subcellular bodies. Although the authors conducted immunofluorescence using anti-HSP40 antibody in SH-SY5Y cells (Extended Data Fig. 1i), which suggests partial colocalization with PCBP2 (as indicated by the yellow signal), this does not fully resolve the initial concern. To strengthen the interpretation, the authors are encouraged to perform additional immunostaining with an OC antibody (anti-amyloid fibril) or Thioflavin S (ThioS) staining, in combination with PCBP2 staining, on post-mortem brain tissue sections and SHSY-5Y cells.
2. Fig. 1h, it is suggested that the authors extend the imaging time course to 20–30 minutes to ensure the complete dissolution of condensates, thereby confirming the absence of any residual entities of condensates. Additionally, providing a time-lapse video of the dissolution process would enhance the clarity and impact of the data presentation.
3. The manuscript frequently refers to "PCBP2-555." To avoid ambiguity, this should be denoted as "PCBP2-A555," indicating Alexa Fluor 555-labeled recombinant PCBP2 protein.
4. The FRAP experiment with recombinant PCBP2 in vitro (Fig. 1j) is commendable. This raises an interesting question about the evolution of material properties of condensates over time, as well as their fusion behavior. Further discussion or data in this regard would be appreciated.
5. It is suggested that the authors perform FRAP experiments in SH-SY5Y cells at early and late time points post-induction (e.g., 24 to 48 hours). This would provide direct insight into the dynamics of PCBP2 condensates in a cellular environment and their potential involvement in disease pathogenesis.
6. The α -CTF band in the revised Extended Data Fig. 3b remains faint and difficult to interpret in terms of its downregulation upon PCBP2 knockdown. A clearer and higher-quality Western blot is recommended to substantiate the claim.
7. The Western blot in Fig. 5b shows approximately 25% reduction in PCBP2 levels after CN0928 treatment. However, Fig. 5c shows a much more substantial signal reduction via live-cell imaging. The authors are requested to provide a plausible explanation for this apparent discrepancy between protein quantification and imaging results.

Additionally, I have gone through carefully all the comments by reviewer 3 and corresponding responses from the authors' end. I strongly believe the authors have done very good job addressing all the points raised by reviewer 3. In the context of Reviewer 3's comments, I think the manuscript looks very good now and is suitable for acceptance."

Reviewer #2

(Remarks to the Author)

In this revised manuscript (previously submitted to [journal name redacted]) the authors included additional data in an effort to address specific points raised by the reviewers. While the manuscript has been improved, there are still a number of important issues that require more careful attention. In my comments below I am specifically addressing the deficiencies identified in my previous review.

1a. The authors now document that the purified PCBP2 can form droplets in the test tube. This is a significant addition. However, these experiments were performed at a high protein concentration (5 μ M) in the presence of very large concentration (10%) of a crowding agent, PEG. Could the authors comment on physiological relevance of these conditions (e.g., how this compares to the physiological concentration of PCBP2)?

1b. Even though the experiments in the test tube indicate that PCBP2 has some propensity for LLPS, the evidence that the granules observed in cells are indeed liquid condensates still remains weak. First, the new data claiming that the granules are dissolved upon 1,6-HD treatment (Fig. 1h) are not very convincing, as the resolution/quality of these images is low, and granular structures seem to be still present after the treatment. Furthermore, while 1,6-HD is known to disassemble some condensates, it is a non-specific agent that could also affect the structure of other types of assemblies such as aggregates. Second, the video included as supplementary data is equally unconvincing and does not support the claim that the granules undergo fusion and fission (not to mention that, contrary to authors' claim, fission is certainly not a hallmark of LLPS). Could the authors perform (as I previously suggested) FRAP experiments on these granules? This would provide a more convincing evidence regarding liquid-like nature of these granules.

2. New data shown in Fig. 2j do not really address to question whether PCBP2 acts as driver of condensate formation. This is because the second protein used in these experiments (TIA1) also undergoes LLPS on its own. Thus, from the design of these experiments one cannot tell which of these two proteins acts a driver of LLPS, and which is just a client.

Minor points:

1. What is the difference between the middle and bottom panels in Fig. 1? This should be indicated in the figure legend.
2. p. 4, line 10 – The word “disseminating” is a bit out of context here. It should be replaced with “dissolving”.
3. p. 4, line 12 – “lipid-like droplets” should be replaced with “liquid-like droplets”.

Reviewer #4

(Remarks to the Author)

Version 1:

Reviewer comments:

Reviewer #1

(Remarks to the Author)

The authors have addressed all of my previous queries. Now manuscript is significantly improved and may be accepted for publication.

Reviewer #2

(Remarks to the Author)

The authors have satisfactorily addressed my major concerns, providing additional experimental data. The revised manuscript is significantly improved and I support its publication.

Reviewer #4

(Remarks to the Author)

REVIEWER COMMENTS

Reviewer #1 (Remarks to the Author):

The revised manuscript by Wang et al. provides a thorough and insightful exploration of a specific pathological pathway in Alzheimer's disease and proposes potential therapeutic interventions. The authors have made a commendable effort to address the concerns raised in the previous review round. Below are a few minor comments on the revised submission. Upon satisfactory responses, I recommend the acceptance of this article.

A: Thank you for acknowledging our efforts to address the previous concerns and for recommending our work for acceptance pending minor revisions. Please refer to the point-by-point responses below with the corresponding changes in the manuscript. We hope these revisions would meet your expectations.

Comments:

1. In Fig. 1b, the authors performed immunostaining of temporal cortex tissue sections from AD patients and non-AD individuals, where PCBP2 (red pseudocolor) indicates protein clustering. The authors were asked to clarify whether these clusters represent biomolecular condensates or amorphous protein aggregates, potentially sequestered in aggresomes or other subcellular bodies. Although the authors conducted immunofluorescence using anti-HSP40 antibody in SH-SY5Y cells (Extended Data Fig. 1i), which suggests partial colocalization with PCBP2 (as indicated by the yellow signal), this does not fully resolve the initial concern. To strengthen the interpretation, the authors are encouraged to perform additional immunostaining with an OC antibody (anti-amyloid fibril) or Thioflavin S (ThioS) staining, in combination with PCBP2 staining, on post-mortem brain tissue sections and SHSY-5Y cells.

A: we apologize for not caching your previous point properly.

We performed additional experiments using Thioflavin S (ThioS) that stains amyloid fibrils (Urbanc et al., 2002). As shown in the figure panels below, a majority of PCBP2 condensates did not colocalize with ThioS-positive clusters in SH-SY5Y cells, brain sections of AD patients and 5×FAD mice. These data are placed in **Extended Data Fig. 1i,k**, with related revisions in the text (marked blue).

2. Fig. 1h, it is suggested that the authors extend the imaging time course to 20–30 minutes to ensure the complete dissolution of condensates, thereby confirming the absence of any residual entities of condensates. Additionally, providing a time-lapse video of the dissolution process would enhance the clarity and impact of the data presentation.

A: We extend the imaging time course to 60 min in SH-SY5Y cells stably expressing mCherry-PCBP2 (SH-SY5Y-mCherry-PCBP2) treated with 1,6-HD. As shown in **revised Fig 1h**, 1,6-HD dramatically reduced the size and number of PCBP2 clusters at 25 min. A complete loss of PCBP2 clusters could be found at 60 min timepoint, whereas cell shrinkage was remarkable, likely due to 1,6-HD cytotoxicity. A representative time-lapse movie of the dissolution process has been included (**Supplementary Video 2**).

3. The manuscript frequently refers to "PCBP2-555." To avoid ambiguity, this should be denoted as "PCBP2-A555," indicating Alexa Fluor 555-labeled recombinant PCBP2 protein.

A: "PCBP2-555" has been replaced with "PCBP2-A555", many thanks.

4. The FRAP experiment with recombinant PCBP2 in vitro (Fig. 1j) is commendable. This raises an interesting question about the evolution of material properties of condensates over time, as well as their fusion behavior. Further discussion or data in this regard would be appreciated.

A: Thank you for raising this interesting issue.

We assessed the properties of recombinant PCBP2 as you have suggested. To combinedly address the concerns of the reviewing expert 2 (Q1), we replaced PEG with RNA (20 ng/μL), and reduced PCBP2 concentration from 5 μM to 2μM in the reaction buffers, these concentrations fall within physiological range (please also refer to "A to Q1" for reviewing expert 2). Accordingly, data regarding phase separation of PCBP2 under these

conditions are added in **revised Fig. 2c,d**.

As shown in the figure below (**Extended Data Fig. 2** in manuscript), nascent droplets appeared within 5 min following incubation, grew and became stable at 10 min, fused at 1 h and 24 h, respectively.

5. It is suggested that the authors perform FRAP experiments in SH-SY5Y cells at early and late time points post-induction (e.g., 24 to 48 hours). This would provide direct insight into the dynamics of PCBP2 condensates in a cellular environment and their potential involvement in disease pathogenesis.

A: PCBP2-mCherry was transiently introduced into SH-SY5Y cells, and FRAP experiments was designed to be performed at 24 h and 48 h post-transfection, respectively. We noticed that mCherry-PCBP2 condensates become evident in SH-SY5Y cells transiently expressing mCherry-PCBP2 for 48 h but not 24 h, in line with the finding that enhancement of PCBP2 protein level is associated with condensate formation (**Extended Data Fig. 1 f-h**).

As shown in the figure below (**revised Fig. 2e**), in the photobleached region (dashed circle) of SH-SY5Y cells transiently expressing mCherry-PCBP2 for 48 h, PCBP2 condensate signals were immediately vanished upon photobleaching, and these signals were recovered along with the time in 60 s.

We also performed FRAP testing in SH-SY5Y cells stably expressing mCherry-PCBP2. As shown in **Extended Data Fig. 2b**, signal bleaching and recovery behavior was found, in a similar manner to that in cells transiently expressing mCherry-PCBP2 for 48 h, indicating that condensate formation occurs at 48 h post-transfection and in stable cell line.

mCherry-PCBP2 condensates are evident in SH-SY5Y cells at 48 h post-transfection but not at 24 h. Representative time-lapse fluorescence images (left) and quantified fluorescence recovery after photobleaching curve (right) illustrate the recovery of mCherry-PCBP2 condensates at 48 h. The dashed box marks the photobleached region of interest (ROI).

6. The α -CTF band in the revised Extended Data Fig. 3b remains faint and difficult to interpret in terms of its downregulation upon PCBP2 knockdown. A clearer and higher-quality Western blot is recommended to substantiate the claim.

A: We repeated the α -CTF Western blotting under optimized conditions, involving a longer electrophoresis time to improve band separation and an increased image exposure to enhance α -CTF signal. The previous image was replaced by a new one shown below (Extended Data Fig. 3b).

Western blots (left) and quantification (right) of α / β -CTF in SH-SY5Y cells by PCBP2 silencing.

7. The Western blot in Fig. 5b shows approximately 25% reduction in PCBP2 levels after CN0928 treatment. However, Fig. 5c shows a much more substantial signal reduction via live-cell imaging. The authors are requested to provide a plausible explanation for this apparent discrepancy between protein quantification and imaging results.

A: Thank you so much for pointing out this discrepancy.

After a careful look into this image, we realized that the fields in original Fig. 5c (revised Fig. 6c) were selected as an extreme situation and therefore not representative. We apologize for this carelessness. Upon re-examining the full imaging dataset, a more representative image set were then placed in CN-0928 group, please refer to Fig. 6c for details.

Additionally, I have gone through carefully all the comments by reviewer 3 and corresponding responses from the authors' end. I strongly believe the authors have done very good job addressing all the points raised by reviewer 3. In the context of Reviewer 3's comments, I think the manuscript looks very good now and is suitable for acceptance."

On behalf of all coauthors, I would like to sincerely express our appreciation for your extra and kind help to this work.

Reviewer #2 (Remarks to the Author):

In this revised manuscript (previously submitted to [journal name redacted]) the authors included additional data in an effort to address specific points raised by the reviewers. While the manuscript has been improved, there are still a number of important issues that require more careful attention. In my comments below I am specifically addressing the deficiencies identified in my previous review.

We truly appreciate your efforts and commitments to the quality improvement of this manuscript. We have collected additional data and provided point-by-point responses that are shown below, and hope that the revisions would satisfactorily address the remaining concerns.

1a. The authors now document that the purified PCBP2 can form droplets in the test tube. This is a significant addition. However, these experiments were performed at a high protein concentration (5 μM) in the presence of very large concentration (10%) of a crowding agent, PEG. Could the authors comment on physiological relevance of these conditions (e.g., how this compares to the physiological concentration of PCBP2)?

A: Thank you for raising this critical issue.

We performed additional experiments using PCBP2 at 0-4 μM , in combination of cellular extracted RNA ranging from 0-50 $\text{ng}/\mu\text{L}$. It seems that PCBP2 starts to form condensate at 2 μM in the presence of RNA at 10 $\text{ng}/\mu\text{L}$. We think that these concentrations should be within physiological range, supported by the following:

(1) intracellular PCBP2 concentration: total cellular protein is about 200 mg/mL , corresponding to 5 mM (Milo, 2013; Wiśniewski et al., 2014). According to PaxDb data (Wang et al., 2015) (<https://pax-db.org/protein/9606/ENSP00000352438>), PCBP2 abundance in human brain is 256-279 ppm (parts per million), which would lead to a PCBP2 concentration of 1.28-1.40 μM under normal condition. This concentration could be increased to 2 μM in the brain of aged mice and SH-SY5Y mCherry-PCBP2 cells (Extended Data Fig. 1f-h), where PCBP2 condensation is favored.

(2) intracellular RNA concentration: It is reported that total RNA concentration is 150 $\text{ng}/\mu\text{L}$ in human brain (Wu et al., 2020), and 50 $\text{ng}/\mu\text{L}$ in cultured human fibroblasts (Zainuddin et al., 2010), respectively.

These data have been incorporated into a new Figure (**revised Figure 2c,d**), with related revisions in the text.

1b. Even though the experiments in the test tube indicate that PCBP2 has some propensity for LLPS, the evidence that the granules observed in cells are indeed liquid condensates still remains weak. First, the new data claiming that the granules are dissolved upon 1,6-HD treatment (Fig. 1h) are not very convincing, as the resolution/quality of these images is low, and granular structures seem to be still present after the treatment. Furthermore, while 1,6-HD is known to disassemble some condensates, it is a non-specific agent that could also affect the structure of other types of assemblies such as aggregates. Second, the video included as supplementary data is equally unconvincing and does not support the claim that the granules undergo fusion and fission (not to mention that, contrary to authors' claim, fission is certainly not a hallmark of LLPS). Could the authors perform (as I previously suggested) FRAP experiments on these granules? This would provide more convincing evidence regarding liquid-like nature of these granules.

A: We apologize for not properly catching your previous point.

To combinedly address this issue with that raised by the reviewing expert 1 (Q5), FRAP experiments was designed to be performed at 24 h and 48 h post-transfection, respectively. We noticed that mCherry-PCBP2 condensates become evident in SH-SY5Y cells transiently expressing mCherry-PCBP2 for 48 h but not 24 h. As shown in the figure below (**revised Fig. 2e**), in the photobleached region (dashed circle) of SH-SY5Y cells transiently expressing mCherry-PCBP2 for 48 h, PCBP2 condensate signals were immediately vanished upon photobleaching, and these signals were recovered along with the time in 60 s.

We also performed FRAP testing in SH-SY5Y cells stably expressing mCherry-PCBP2. As shown in **Extended Data Fig. 2b**, signal bleaching and recovery behavior was found, in a similar manner to that in cells transiently expressing mCherry-PCBP2 for 48 h, indicating that condensate formation occurs at 48 h post-transfection and in stable cell line.

Also, a high-resolution image regarding PCBP2 clusters in 1,6-HD-treated SH-SY5Y-mCherry-PCBP2 cells were placed as new **Fig. 1h**. We extended assessing time for up to 60 min, based on the advice by the reviewing expert 1 (A to Q2). 1,6-HD dramatically reduced the size and number of PCBP2 clusters at 25 min. A complete loss of PCBP2 clusters could be found at 60 min timepoint when cell shrinkage was remarkable, likely due to 1,6-HD cytotoxicity. An additional representative time-lapse movie of the dissolution process has been included as **Supplementary Video 2**.

2. New data shown in Fig. 2j do not really address to question whether PCBP2 acts as driver of condensate formation. This is because the second protein used in these experiments (TIA1) also undergoes LLPS on its own. Thus, from the design of these experiments one cannot tell which of these two proteins acts a driver of LLPS, and which is just a client.

A: We thank you for pointing out that using TIA1 (itself LLPS-competent) cannot establish PCBP2 as a driver. To address this, we selected TOM20 as a non-driving partner under our assay conditions. To the best of our knowledge, TOM20, one of the PCBP2 condensate components and also mitochondrial proteins, is not reported to show phase separation. Protein sequence analysis (uniprot number Q15388) reveals that TOM20 protein does not contain any known IDR. Thus, we selected TOM20 for further assessment.

(1) Estimating physiological TOM20 protein concentration from PaxDb (Wang et al., 2015). The abundance values of 44-138 ppm correspond to 0.22-0.69 μM of TOM20 protein; we therefore tested TOM20 at 0.5 μM . (2) different combination of purified TOM20 (TOM20-D488 at 0.5 or 1 μM) with 10% PEG or 50 ng/ μL RNA did not lead to a droplet formation. (3) co-incubation of TOM20-D488 (0.5 μM) with PCBP2 (PCBP2-A555, 2 μM) in the presence of RNA (20 ng/ μL) yielded a clear liquid-like droplet. These results support

PCBP2 as the driver in this mixture, whereas TOM20 may function as a client protein.

The figure below is incorporated into **Fig. 3j**, with corresponding revisions in the text (marked blue).

Minor points:

1. What is the difference between the middle and bottom panels in Fig. 1? This should be indicated in the figure legend.

A: We assume this point is about Fig.1i (**revised Fig. 2a**), in which the middle panel shows an isolated single small condensate, whereas the bottom panel shows a cluster of small condensates. The related description has been added to the figure legend.

Other details in figure legends were included as possible as we can. Thanks a lot.

2. p. 4, line 10 – The word “disseminating” is a bit out of context here. It should be replaced with “dissolving”.

A: The word “disseminating” has been replaced with “dissolving” (**p. 4, line 14**).

3. p. 4, line 12 -“lipid-like droplets” should be replaced with “liquid-like droplets”.

A: We have replaced “lipid-like droplets” with “liquid-like droplets”. Thank you once again for your valuable feedback, which has undoubtedly improved the rigor of our study.

Reviewer #4 (Remarks to the Author):

References:

Milo, R. (2013). What is the total number of protein molecules per cell volume? A call to rethink some published values. *BioEssays : news and reviews in molecular, cellular and developmental biology* 35, 1050-1055.

Urbanc, B., Cruz, L., Le, R., Sanders, J., Ashe, K.H., Duff, K., Stanley, H.E., Irizarry, M.C., and Hyman, B.T. (2002). Neurotoxic effects of thioflavin S-positive amyloid deposits in transgenic mice and Alzheimer's disease. *Proceedings of the National Academy of Sciences of the United States of America* 99, 13990-13995.

Wang, M., Herrmann, C.J., Simonovic, M., Szklarczyk, D., and von Mering, C. (2015). Version 4.0 of PaxDb: Protein abundance data, integrated across model organisms, tissues, and cell-lines. *Proteomics* 15, 3163-3168.

Wiśniewski, J.R., Hein, M.Y., Cox, J., and Mann, M. (2014). A "proteomic ruler" for protein copy number and concentration estimation without spike-in standards. *Molecular & cellular proteomics : MCP* 13, 3497-3506.

Wu, L.R., Fang, J.Z., Khodakov, D., and Zhang, D.Y. (2020). Nucleic Acid Quantitation with Log-Linear Response Hybridization Probe Sets. *ACS sensors* 5, 1604-1614.

Zainuddin, A., Chua, K.H., Abdul Rahim, N., and Makpol, S. (2010). Effect of experimental treatment on GAPDH mRNA expression as a housekeeping gene in human diploid fibroblasts. *BMC molecular biology* 11, 59.